# DRAXIN regulates interhemispheric fissure remodelling to influence the extent of corpus callosum formation

Laura Morcom[1†‡], Timothy J Edwards[1,2†§], Eric Rider[3], Dorothy Jones-Davis[3], Jonathan WC Lim[1], Kok-Siong Chen[1#], Ryan J Dean[1], Jens Bunt[1¶], Yunan Ye[1], Ilan Gobius[1**], Rodrigo Suárez[1], Simone Mandelstam[4], Elliott H Sherr[3*], Linda J Richards[1,5*]

[1]The University of Queensland, Queensland Brain Institute, Brisbane, Australia; [2]Faculty of Medicine, Brisbane, Australia; [3]Departments of Neurology and Pediatrics, Institute of Human Genetics and Weill Institute of Neurosciences, University of California, San Francisco, San Francisco, United States; [4]Department of Radiology, University of Melbourne, Royal Children's Hospital, Parkville, Australia; [5]School of Biomedical Sciences, Brisbane, Australia

*For correspondence:
Elliott.Sherr@ucsf.edu (EHS);
richards@uq.edu.au (LJR)

[†]These authors contributed equally to this work

Present address: [‡]The University of Cambridge, Department of Paediatrics, Wellcome-MRC Stem Cell Institute, Cambridge, United Kingdom; [§]Great Ormond Street Institute of Child Health, University College London, London, United Kingdom; [#]Harvard Medical School, Brigham and Women's Hospital, Boston, United States; [¶]Princess Maxima Centre for Pediatric Oncology, Utrecht, Netherlands; [**]The University of Quensland, Diamantina Institute, Brisbane, Australia

**Abstract** Corpus callosum dysgenesis (CCD) is a congenital disorder that incorporates either partial or complete absence of the largest cerebral commissure. Remodelling of the interhemispheric fissure (IHF) provides a substrate for callosal axons to cross between hemispheres, and its failure is the main cause of complete CCD. However, it is unclear whether defects in this process could give rise to the heterogeneity of expressivity and phenotypes seen in human cases of CCD. We identify incomplete IHF remodelling as the key structural correlate for the range of callosal abnormalities in inbred and outcrossed BTBR mouse strains, as well as in humans with partial CCD. We identify an eight base-pair deletion in *Draxin* and misregulated astroglial and leptomeningeal proliferation as genetic and cellular factors for variable IHF remodelling and CCD in BTBR strains. These findings support a model where genetic events determine corpus callosum structure by influencing leptomeningeal-astroglial interactions at the IHF.

## Introduction

The corpus callosum (CC) is the largest white matter tract that mediates information transfer between brain hemispheres in placental mammals. In humans, it fails to form normally in approximately 1:4000 live births, resulting in a group of conditions collectively termed CC dysgenesis (CCD; *Glass et al., 2008*). CCD incorporates complete and partial congenital absence of the CC, as well as hypo- and hyperplasia (thinning or thickening, respectively) of the CC. Each of these structural phenotypes can variably impact brain function and organisation (*Brown and Paul, 2019*; *Edwards et al., 2014*; *Paul et al., 2007*), but the precise developmental mechanisms that could give rise to these diverse CCD phenotypes remain unknown.

CC formation is dependent on a prior sequence of developmental processes: cellular proliferation, migration, axonal elongation, guidance, and targeting (*Donahoo and Richards, 2009*; *Edwards et al., 2014*; *Morcom et al., 2015*). Callosal axons derived from cells within the cingulate and neocortices elongate and cross the telencephalic midline in a region of the septum termed the commissural plate (*Moldrich et al., 2010*; *Rakic and Yakovlev, 1968*). We previously demonstrated that in order for callosal axons to cross the midline, the interhemispheric fissure (IHF) that separates the telencephalic hemispheres must be remodelled to form a permissive substrate (*Gobius et al.,*

2016). IHF remodelling is mediated by intercalation of specialised astroglia, known as the midline zipper glia (MZG), across the IHF (*Silver et al., 1993*; *Gobius et al., 2016*). This process does not occur in naturally acallosal mammalian marsupial and monotreme species (*Gobius et al., 2016*), and its failure appears to be a major cause of complete CCD in humans (*Gobius et al., 2016*). When IHF remodelling does not proceed normally in placental mammals, callosal axons do not cross the midline and can instead form longitudinal tracts in the ipsilateral hemisphere that are known as Probst bundles (*Probst, 1901*). Although IHF remodelling is a prerequisite for CC formation, our understanding of the cellular and genetic factors involved is incomplete. Moreover, it is unknown whether disruptions to this process could account for the spectrum of commissural phenotypes seen in CCD or whether these are due to independent developmental mechanisms.

Absence or dysgenesis of the hippocampal commissure (HC) is frequently observed to co-occur with CCD in humans, indicating that these commissures rely on common developmental programs (*Hetts et al., 2006*). Analogously, the BTBR T + Itpr3 tf/J (BTBR) mouse has a severe commissural phenotype incorporating complete CCD and HC dysgenesis (*Wahlsten et al., 2003*). Using the F2 generation of BTBR × C57Bl/6J (C57) intercross, which displays variable CCD and HC dysgenesis, we previously demonstrated that a highly penetrant locus on chromosome 4 was associated with CC and HC size (*Jones-Davis et al., 2013*). This suggests that the degree of dysgenesis of two major telencephalic commissures may result from disruption of a single developmental process. Candidate gene analysis based on filtering for variants predicted to affect functionally relevant genes identified *Draxin* as a favourable gene candidate that might underlie variable CC and HC formation (*Jones-Davis et al., 2013*). *Draxin* encodes a known ligand to the axon guidance receptor DCC and is required for CC and HC formation in mice (*Ahmed et al., 2011*; *Hossain et al., 2013*; *Islam et al., 2009*).

Here, we investigate the underlying genetic and developmental mechanisms leading to diverse CCD phenotypes in BTBR mice and the BTBR × C57 cross. CCD severity and the extent of HC dysgenesis in these mouse strains is strongly associated with abnormal retention of the IHF, and thus the degree to which IHF remodelling is incomplete. Moreover, we describe an eight base-pair deletion in *Draxin*, which truncates and ablates normal DRAXIN protein expression in BTBR mice. Inheritance of the *Draxin* mutation is a driver of defective IHF remodelling and subsequent CCD and HC dysgenesis in BTBR and BTBR × C57 mice. Mis-regulated cellular proliferation of MZG and leptomeningeal cells were both identified as the cellular correlates for failed MZG-mediated IHF remodelling and interhemispheric tract formation in BTBR mice. Finally, we identify incomplete IHF remodelling in a cohort of human individuals with partial CCD. Together, our results suggest that diverse CCD phenotypes can arise from a single genetic event that variably disrupts IHF remodelling and, consequently, the amount of substrate available for CC and HC axons to cross the midline, and therefore provides the first aetiology associated with partial CCD.

## Results

### The CC and HC are variably malformed in BTBR × C57 N2 mice

We previously demonstrated that BTBR × C57 N2 littermates display a spectrum of CCD phenotypes (*Jones-Davis et al., 2013*; *Edwards et al., 2020*). Here, we further classified these phenotypes into full CC (CC length ≥3 mm, according to typical C57 wildtype CC lengths in *Jones-Davis et al., 2013*; *Figure 1A, B*), partial CCD (CC length >0 and <3 mm), and complete CCD (CC length = 0 mm; *Figure 1A, B*). We identified variable HC size in animals with complete CCD, suggesting that distinct subpopulations of CCD with variable HC dysgenesis may occur within the BTBR × C57 N2 mouse (*Figure 1C, D*). Furthermore, the severity of CCD was correlated with HC dysgenesis; HC length was reduced in complete CCD compared to full CC littermates, and was reduced in complete CCD compared to partial CCD (*Figure 1C* and *Supplementary file 1*). These results demonstrate that the BTBR × C57 N2 mice display a range of CC phenotypes suitable for further investigation of underlying aetiologies. Moreover, the association between CC length and HC length (as observed in our initial quantitative trait locus (QTL)-based manuscript on CC and HC morphology; *Jones-Davis et al., 2013*) suggests that a common aetiology may underlie the observed variance in commissural size in this mouse cross.

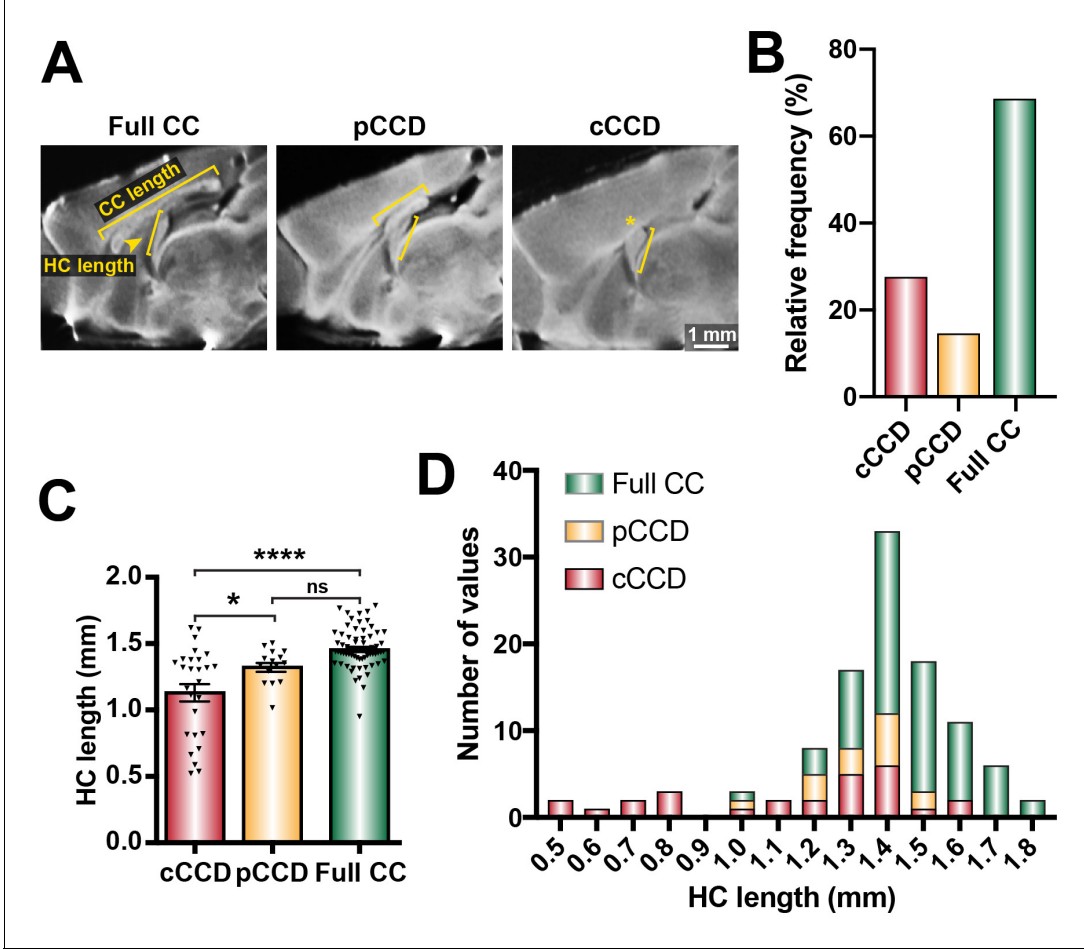

**Figure 1.** Distribution of commissural size in BTBR × C57 N2 mice. (**A**) Corpus callosum (CC) length and hippocampal commissure (HC) length were measured on single-diffusion direction MRI scans for n = 112 adult BTBR × C57 N2 mice. (**B**) The relative frequency of three distinct subsets of CC phenotypes based on CC length: complete corpus callosum dysgenesis (cCCD; red), partial CCD (pCCD; yellow), and normal CC length (Full CC; green). (**C**) Group-wise comparison between callosal phenotypes for HC length. (**D**) Stacked histogram of HC length for each callosal phenotype. Data represented as mean ± SEM. *p<0.05; ****p<0.0001, ns = not significant, as determined by one-way ANOVA with post Tukey's multiple comparisons test.

The online version of this article includes the following source data for figure 1:

**Source data 1.** Distribution of commissural size in BTBR × C57 N2 mice.

## CC and HC malformations are associated with defects in IHF remodelling in the BTBR × C57 N2 and BTBR parental mouse strain

To investigate potential structural correlates of complete and partial CCD, we first examined the midline of BTBR and BTBR × C57 N2 mice. We previously demonstrated that remodelling of the IHF is a critical developmental step required for subsequent midline crossing of callosal axons, and that failure of this process to occur results in an unfused septum; an MRI feature that is strongly associated with complete CCD in humans (*Gobius et al., 2016*). To determine whether incomplete IHF remodelling may underlie the spectrum of CCD seen in the BTBR × C57 N2 mouse, we acquired high-resolution structural MRI scans of adult C57 and complete CCD BTBR mice, as well as a subset of BTBR × C57 N2 mice with complete and partial CCD.

C57 adult mice demonstrated a fused septum and IHF positioned anteriorly and superiorly to the CC and HC (*Figure 2A*). In contrast, the acallosal BTBR adult mouse demonstrated an unfused septum and an IHF that is aberrantly retained across almost the full extent of the telencephalic midline (*Figure 2B*, yellow bracket). Consistent with what has been shown previously, the only crossing white matter identified in the dorsal telencephalon of the BTBR mouse was a small, ventrally positioned

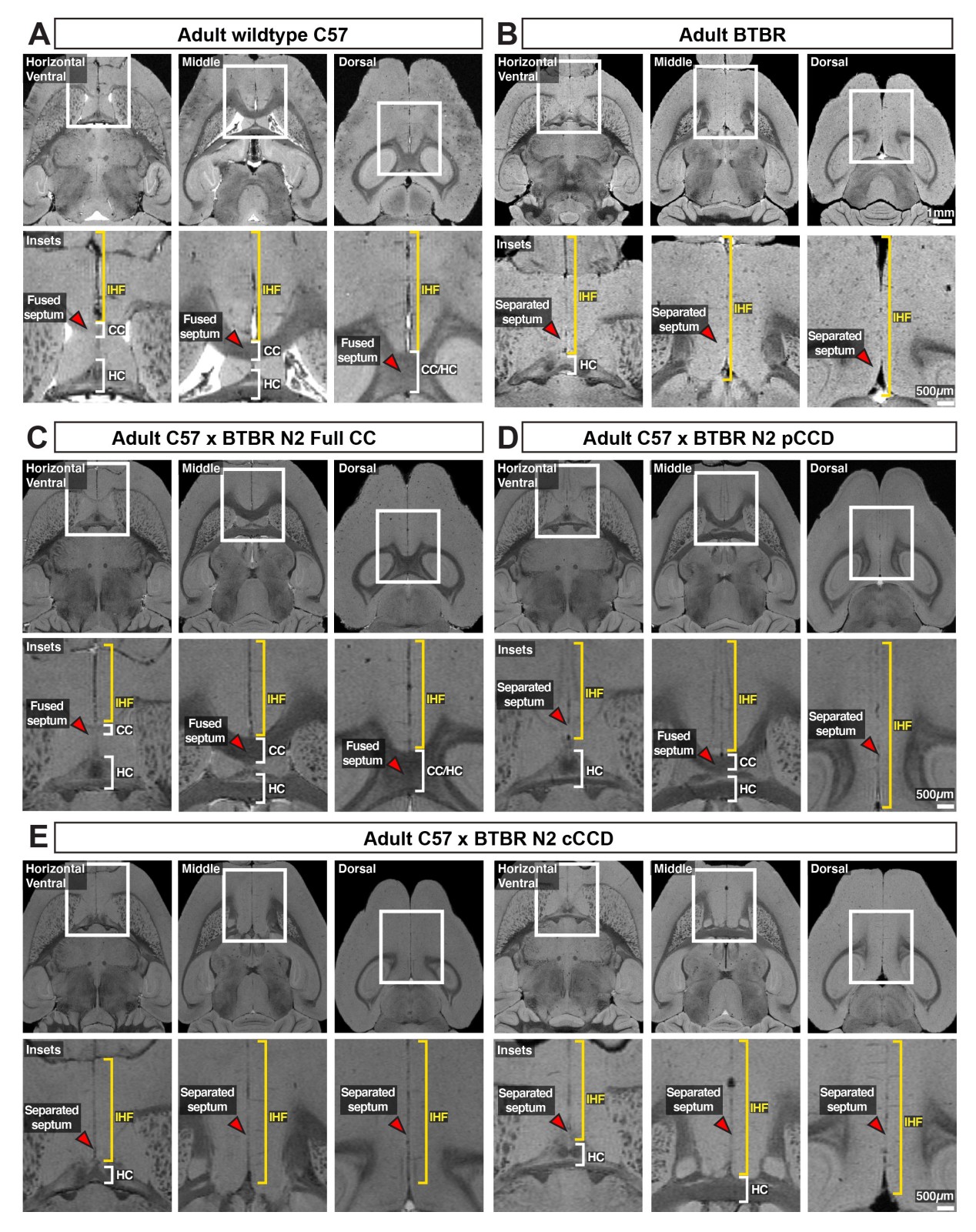

**Figure 2.** Axial structural MRI slices and insets (white rectangles) of telencephalic midline anatomy in adult wildtype. C57 (**A**) and acallosal BTBR parental mice (**B**), as well as adult BTBR × C57 N2 mice with distinct commissural phenotypes (**C–E**). The interhemispheric fissure (IHF) is indicated with yellow brackets, the corpus callosum (CC) and hippocampal commissure are indicated with white brackets, and the septum is indicated with red

*Figure 2 continued on next page*

*Figure 2 continued*

arrowheads. n = 3 for C57 and BTBR parental strains, n = 10 for each callosal condition for the BTBR × C57 N2 cross. cCCD: complete corpus callosum dysgenesis; pCCD: partial corpus callosum dysgenesis.

HC (*Wahlsten et al., 2003*). Full CC BTBR × C57 N2 mice demonstrated similar midline anatomy to the C57 mice (*Figure 2C*). Partial CCD mice demonstrated an intact HC; however, the anterior-posterior extent of the CC was reduced and was associated with dorsal retention of the IHF, suggesting that the IHF had not been fully remodelled (*Figure 2D*). In support of an aetiological link between an IHF remodelling defect and CCD severity, complete CCD mice had a more severe IHF remodelling defect, with near complete retention of the IHF (*Figure 2E*). A subset of complete CCD mice displayed a more severe IHF phenotype with an associated dysgenesis of the HC that recapitulates that seen in the BTBR parental strain (*Figure 2E*, left panels) compared to complete CCD mice with an intact HC (*Figure 2E*, right panels). Together, these data suggest that the underlying pathogenesis of CCD in the BTBR × C57 N2 mouse is incomplete IHF remodelling, leading to aberrant retention of the IHF. This further suggests that the severity of the HC and callosal phenotypes is related to the degree of IHF remodelling that occurs.

## Retention of the IHF with an unfused or absent septum is associated with the degree of partial CCD in humans

An unfused septum and deep IHF are invariably present in humans with complete CCD, who commonly have associated HC malformations (*Gobius et al., 2016*; *Hetts et al., 2006*). To determine whether an analogous relationship between septal and IHF abnormalities and CCD severity to that seen in the BTBR × C57 N2 mouse might exist in humans with partial CCD irrespective of genetic cause, we examined the septum and IHF in structural MRI scans of 10 adult individuals with partial CCD and compared these to 9 neurotypical controls (*Figure 3A*, *Figure 3—figure supplement 1*). Partial CCD individuals demonstrated variably positioned CC remnants comprising one or more, but not all, of the normal CC segments. These individuals often had other mild brain abnormalities that were deemed to be not related to midline formation and IHF remodelling, except for two individuals that demonstrated interhemispheric cysts.

In neurotypical individuals, the CC extends dorsal, anterior, and posterior to the septum, which is fully fused except in a minority of individuals that demonstrate cavum septum pellucidum – a normal anatomical variant that forms ventral to the corpus callosum (*Schwidde, 1952*). Partial CCD individuals demonstrated significantly increased IHF length in the dorsoventral and anterior-posterior axis (*Figure 3A–D*, *Figure 3—figure supplement 1B, C*, and *Supplementary file 1*), and a significant reduction in the length of fused septum along the anterior-posterior axis as normalised to total midline length (*Figure 3—figure supplement 1D* and *Supplementary file 1*). All partial CCD individuals displayed increased length of the posterior IHF in the axial plane outside of the range measured in neurotypical individuals (*Figure 3H*). Increased length of the anterior and posterior IHF indicates a decrease in IHF remodelling and septal fusion, and therefore a reduced amount of tissue available for callosal midline crossing. This correlated with a reduction in CC width in partial CCD; the posterior CC being the most severely affected since it was absent in all but 1 individual with partial CCD (*Figure 3I, J*). Notably, the CC remnant in partial CCD individuals was often displaced ventrally within the fused septum where it is not normally evident in neurotypical controls (*Figure 3C* and *Figure 3—figure supplement 1B*). Several individuals displayed absence of the septal leaves that occurred in association with an interhemispheric cyst (partial CCD subjects 6 and 10; *Figure 3* and *Figure 3—figure supplement 1*). These results suggest that partial CCD is associated with incomplete IHF remodelling and septal fusion in our cohort. Therefore, developmental failure of IHF remodelling appears to be consistently associated with a spectrum of CCD phenotypes in mice and humans.

## A deletion within *Draxin* in the parental BTBR strain is associated with loss of *Draxin* expression at the telencephalic midline

Families with CCD can exhibit variable expressivity when carrying the same inherited pathogenic gene variant (*Marsh et al., 2017*). Because complete and partial CCD BTBR × C57 N2 littermates

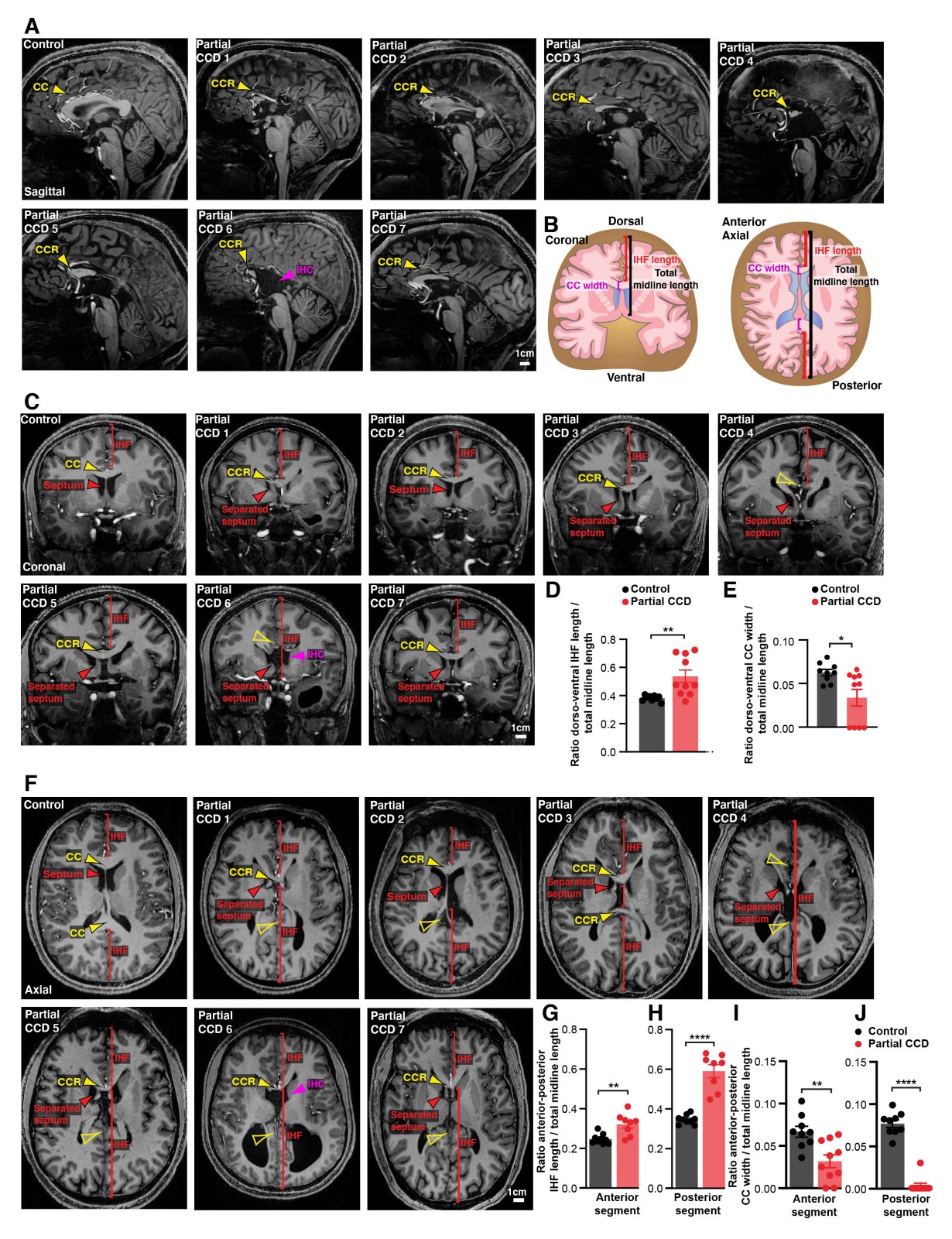

**Figure 3.** Structural MRI study of interhemispheric fissure (IHF) and septal defects associated with partial corpus callosum dysgenesis (CCD) in humans. Representative sagittal (**A**), coronal (**C**), and axial (**F**) slices from T1-weighted structural scans on 9 neurotypical (control) individuals and 10 individuals with partial CCD. The CC or the CC remnant (CCR) is indicated with yellow arrowheads, interhemispheric cysts (IHC) are indicated with magenta arrowheads, the IHF extent is indicated with red brackets, and the septum is indicated with red arrowheads. The IHF length and CC width were

*Figure 3 continued on next page*

*Figure 3 continued*

measured from coronal and axial images and normalised to the length of the midline, as quantified in (D), (E), (G), (H), (I), (J) and schematised in (B). Data is represented as mean ± SEM. *p<0.05; **p<0.01, ****p<0.0001 as determined with unpaired t tests or Mann–Whitney tests. See *Figure 3— figure supplement 1* for further subjects and quantification.

The online version of this article includes the following source data and figure supplement(s) for figure 3:

**Source data 1.** Ratio of anterior-posterior interhemispheric fissure (IHF) length over total midline length in control and partial corpus callosum dysgenesis (pCCD) humans.

**Figure supplement 1.** Structural MRI study of interhemispheric fissure (IHF) and septal defects associated with partial corpus callosum dysgenesis (CCD) in humans.

**Figure supplement 1—source data 1.** Ratio of fused septum length over total midline length in control and partial corpus callosum dysgenesis (pCCD) humans.

---

both display an IHF remodelling defect, the severity of which is associated with the severity of CCD, we sought to determine whether they share a genetic aetiology. Our previously described linkage analysis identified a QTL at the distal end of chromosome 4 which demonstrated a high logarithm of the odds (LOD) score for CC and HC anatomy (*Jones-Davis et al., 2013*). Variant filtering of exome sequencing within this QTL identified an eight base-pair deletion introducing a premature stop codon in exon 2 of *Draxin* in the BTBR strain (*Jones-Davis et al., 2013*; *Figure 4A, B*). *Draxin* is a promising candidate to explain CCD in the BTBR × C57 N2 mouse since *Draxin* knockout mice display CCD (*Ahmed et al., 2011*; *Islam et al., 2009*). Moreover, mutations in the gene encoding the DRAXIN receptor, DCC, are also associated with CCD in mice and humans (*Fazeli et al., 1997*; *Finger et al., 2002*; *Fothergill et al., 2014*; *Jamuar et al., 2017*; *Marsh et al., 2018*; *Marsh et al., 2017*). We generated in situ riboprobes for wildtype and mutant *Draxin* using mRNA isolated from C57 and BTBR mice, respectively, and examined the mRNA expression in the BTBR × C57 N2 and the BTBR parental mice. In situ hybridisation with 3' *Draxin* antisense probes from both strains revealed that *Draxin* is highly expressed in the cingulate cortex and the septum in mid-horizontal sections of the telencephalic midline of C57 mice, consistent with previous findings (*Figure 4C*; *Islam et al., 2009*). *Draxin* mRNA was expressed in both C57 mice and mice homozygous for the *Draxin* mutation (BTBR and BTBR × C57 N2 mice), but protein expression was undetectable in tissue using a polyclonal antibody generated from an immunogen of whole human DRAXIN via immunohistochemistry and western blot (*Figure 4C, D*). However, western blot revealed that expression of the BTBR *Draxin* coding sequence in HEK293T cells produced DRAXIN protein of reduced molecular weight compared to the C57 *Draxin* coding sequence (*Figure 4D*). These results suggest that the eight base-pair deletion in *Draxin* truncates DRAXIN (*Figure 4E*) and disrupts normal protein expression in vivo.

## The *Draxin* deletion is associated with complete and partial CCD in BTBR × C57 N2 mice

To further demonstrate linkage of the eight base-pair deletion in *Draxin* to the observed CCD in BTBR and BTBR × C57 N2 mice, we performed Sanger sequencing of a single-nucleotide polymorphism (SNP) on chromosome 4 (rs6397070), 7.795 megabases downstream of the *Draxin* deletion as a marker for the BTBR allele in a subset of complete CCD, partial CCD, and normal CC littermates (n ~ 10 of each phenotype, 31 total). Additional SNPs were also sequenced at candidate loci on chromosome 9 (rs29890894) and chromosome 15 (rs31781085), which were previously judged to have a potential genetic influence on commissure size based on LOD scores that did not reach statistical significance (*Jones-Davis et al., 2013*). We compared commissure length to the allele composition at the chromosome 4 locus (rs6397070) and found that CC length was significantly reduced in mice homozygous for the BTBR allele (C/C) compared to heterozygous mice (C/T; *Figure 5A* and *Supplementary file 1*). In contrast, neither HC length (*Figure 5B*) nor anterior commissure area (*Figure 5C*) demonstrated a significant difference between homozygous and heterozygous mice (*Supplementary file 1*). Of the 21 homozygous mice, 19 had CCD (n = 10 complete CCD, n = 9 partial CCD) and 2 had a normal CC. Of the 10 mice heterozygous at the rs6397070 allele, 9 had a normal CC and 1 mouse had partial CCD. No significant differences between genotypes were identified for chromosome 9 (*Figure 5E–G* and *Supplementary file 1*) or chromosome 15 (*Figure 5I–K* and *Supplementary file 1*) candidate loci for any commissure.

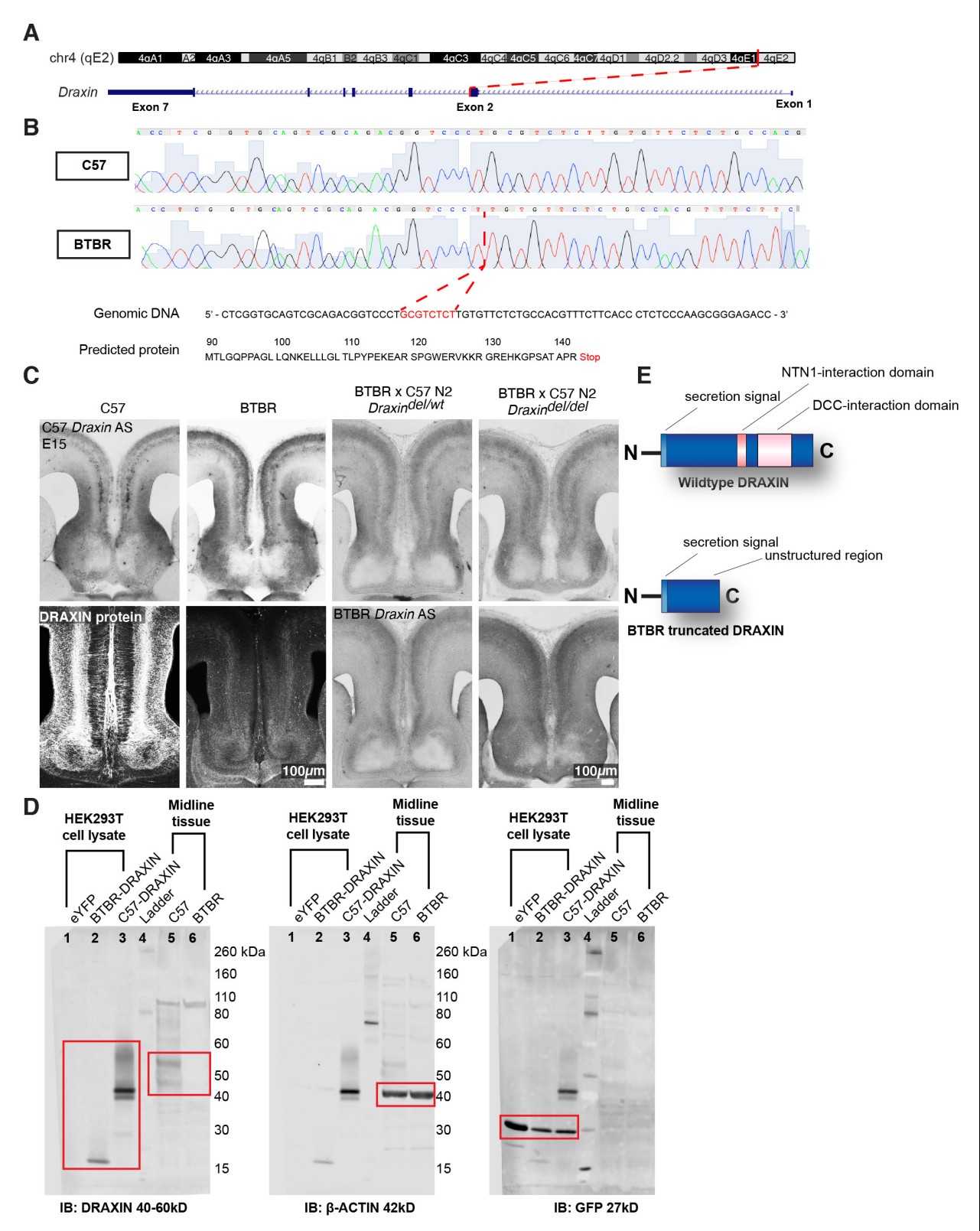

**Figure 4.** An eight base-pair deletion in *Draxin* in BTBR strains truncates and reduces DRAXIN protein expression. Exome sequencing of a candidate region on mouse chromosome 4 (**A**) and confirmatory Sanger sequencing found an eight base-pair deletion in *Draxin*, which introduces a premature stop codon (**B**), predicted to truncate DRAXIN protein (**B, E**). (**C**) In situ hybridisation against 3' C57 or BTBR *Draxin* demonstrates a similar pattern of *Draxin* mRNA expression in C57 and BTBR parental strains and in the BTBR × C57 N2 mice. Fluorescent immunohistochemistry for DRAXIN protein

*Figure 4 continued on next page*

*Figure 4 continued*

(bottom-left panels) demonstrates that DRAXIN is highly expressed in C57 mice at E15 but not in BTBR mice. (D) Cell lysates derived from HEK293T cells expressing pCag-eYFP or pCag-iresGFP with either BTBR or C57 *Draxin* coding sequences were incubated with anti-DRAXIN, anti- β-ACTIN, and anti-GFP antibodies. Specific bands at ~18 kD and ~40–45 kD are shown for DRAXIN (red boxes) and demonstrate that BTBR *Draxin* produces a protein of reduced molecular weight, indicating truncation. Midline tissue lysates from E15 C57 and BTBR mice incubated with anti-DRAXIN and anti-β-ACTIN antibodies reveal (red boxes) specific bands at ~40–60 kD and ~42 kD, respectively, indicating that DRAXIN expression is severely reduced in BTBR mice.

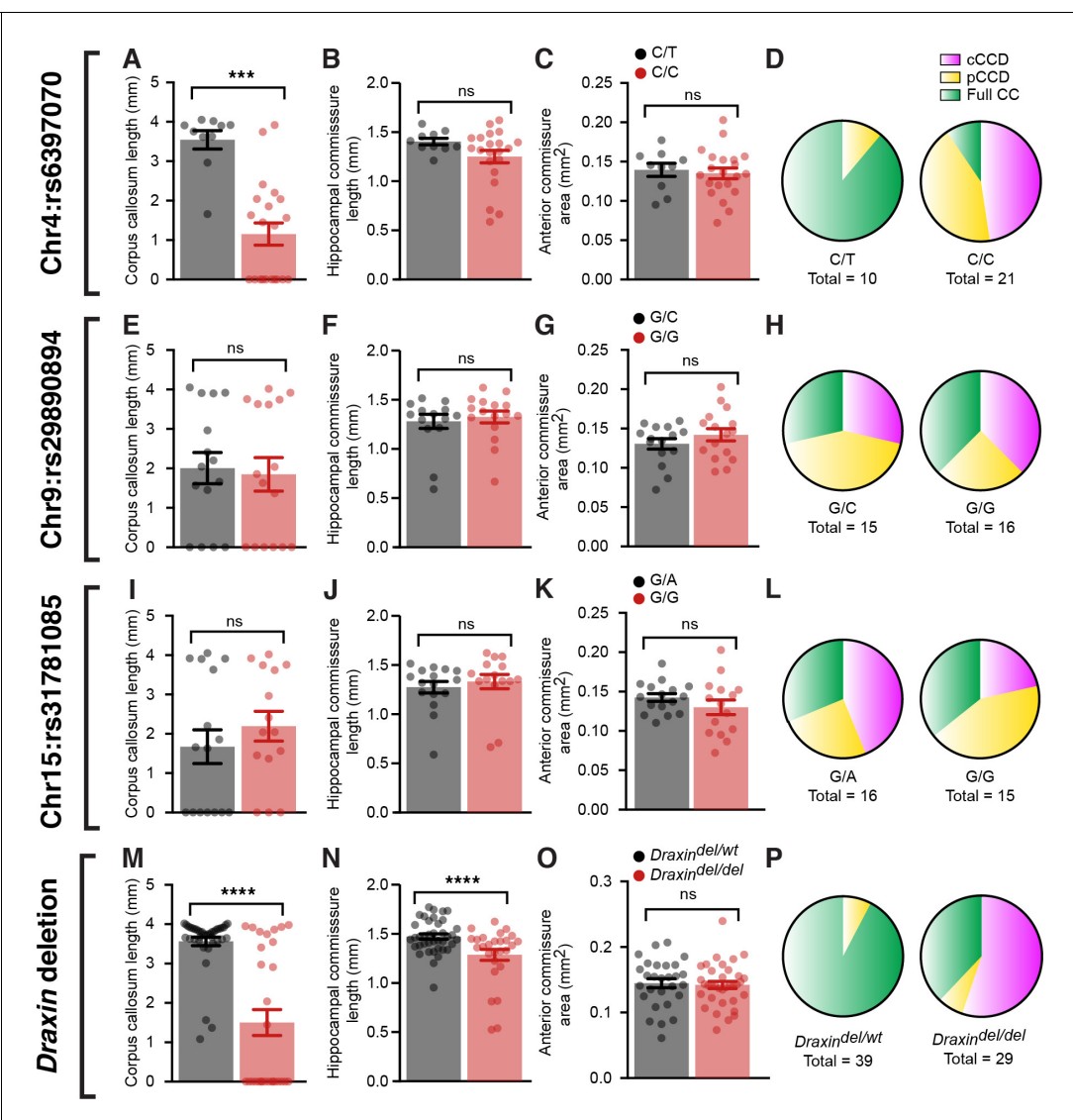

**Figure 5.** BTBR allele on chromosome 4 and a deletion in *Draxin* are associated with corpus callosum dysgenesis (CCD) in BTBR × C57 N2 mice. Group-wise comparison for marker single-nucleotide polymorphisms (SNPs) at candidate genetic loci compared with CC length (A, E, I), hippocampal commissure (HC) length (B, F, J), anterior commissure (AC) area (C, G, K), or frequency of CC phenotypes (D, H, L) in BTBR × C57 N2 mice. Genotyping for the eight base-pair deletion in *Draxin* in BTBR × C57 N2 mice compared with CC length (M), HC length (N), AC area (O), or frequency of CC phenotypes (P). Mice homozygous for the BTBR allele (C/C at SNP rs6397070 on chromosome [Chr] 4) or the *Draxin* deletion have significantly reduced CC length and HC length compared to heterozygous littermates. Data is represented as mean ± SEM. ***p<0.001, ****p<0.0001, ns = not significant, as determined by Kruskal–Wallis ANOVA tests.

The online version of this article includes the following source data for figure 5:

**Source data 1.** Commissure size in BTBR N2 mice based on *Draxin* mutant allele genotype.

To further validate that the *Draxin* mutation is a likely cause of CCD in BTBR × C57 N2 mice, we genotyped for the eight base-pair *Draxin* deletion and examined the influence of the mutation on commissure size. CC length was significantly reduced in BTBR × C57 N2 mice homozygous for the *Draxin* deletion (*Figure 5M* and *Supplementary file 1*), consistent with our results of the candidate SNP on chromosome 4. All 16 mice from a cohort of 68 mice that displayed complete CCD were homozygous for the *Draxin* deletion (*Figure 5P* and *Supplementary file 1*). Of the 47 mice with normal CC length, 11 were homozygous for the *Draxin* deletion, suggesting incomplete penetrance (*Figure 5P* and *Supplementary file 1*), which is also observed in *Draxin* knockout mice (*Hossain et al., 2013*; *Islam et al., 2009*). Contrary to the earlier SNP analysis, we found in this larger, unbiased sample that while anterior commissure area was not influenced by *Draxin* deletion (*Figure 5O* and *Supplementary file 1*), *Draxin* deletion was associated with reduced HC length (*Figure 5N* and *Supplementary file 1*). Together, these results are consistent with the hypothesis that *Draxin* deletion is the primary genetic aetiology underlying failure for IHF remodelling, which results in CCD and HC dysgenesis in the BTBR × C57 N2 mouse.

## Increased somal translocation of MZG and failure to remodel the IHF are associated with CCD in the BTBR strain

To investigate how DRAXIN regulates CC and HC formation in these mice, we first examined cell-type-specific *Draxin* expression in wildtype CD1 mice. In situ hybridisation for *Draxin* with immunohistochemistry for glial markers and components of the IHF was performed prior to the onset of IHF remodelling. *Draxin* mRNA was highly expressed in GLAST-positive radial MZG progenitors in the telencephalic hinge from E12 and was further observed in GLAST-positive MZG migrating to the pan-LAMININ-positive IHF surface at E15 (*Figure 6B, E*). Immunohistochemistry on wildtype E15 horizontal sections revealed that DRAXIN was widely localised within the telencephalic midline on GLAST-positive radial MZG membranes including migrating cells and progenitors at all stages (*Figure 6C, F, K*). DRAXIN was also localised to commissural axons at E15 (*Figure 6H*) and on the basement membrane and leptomeningeal cells within the IHF (*Figure 6H'*). Thus, DRAXIN, which is known to be secreted (*Islam et al., 2009*), is expressed within MZG cells and associates with multiple cellular components of the interhemispheric midline, such that it could regulate the development of MZG, leptomeninges and axons, and the interactions between these cellular populations during IHF remodelling and corpus callosum formation.

To further investigate the function of DRAXIN in CC formation, we probed for cellular phenotypes that may explain the loss of IHF remodelling and CC formation in the BTBR parental strain, which is homozygous for the *Draxin* mutation. Immunohistochemistry for the leptomeningeal marker pan-LAMININ (*Gobius et al., 2016*), axonal marker GAP43, and mature astroglial marker GFAP in E17 inbred BTBR and wildtype C57 mice revealed that BTBR inbred mice display complete retention of the IHF, which manifests as a significantly higher ratio of IHF length to total midline length in BTBR mice compared to wildtype C57 mice (*Figure 7A–C* and *Supplementary file 1*). GFAP-positive MZG remain in two columns of cells that do not intercalate across the IHF in BTBR inbred mice (*Figure 7A*), suggesting that a defect in the astroglial-IHF interaction results in the failure of midline crossing of CC and HC axons in BTBR mice.

The endfeet of radial MZG progenitors normally form attachments to both the third ventricle and IHF within a region known as the telencephalic hinge. They then proliferate and undergo somal translocation to the IHF between E12 and E15 to initiate IHF remodelling at E15 in mice (*Gobius et al., 2016*). Immunohistochemistry for radial MZG using NESTIN and GLAST prior to IHF remodelling revealed that while radial MZG are present between the IHF pial surface and the third ventricle in BTBR mice (*Figure 7D, F*), radial NESTIN-positive MZG fibres were disorganised, and significantly more abundant lateral to the IHF between 100–200 μm from the base of the IHF in BTBR mice (*Figure 7F, G* and *Supplementary file 1*). Moreover, altered radial MZG distribution in BTBR mice was associated with abnormal widening of the base of the IHF (*Figure 7D, E*, *Figure 8—figure supplement 1*, and *Supplementary file 1*); a region that normally undergoes selective compression prior to IHF remodelling as increasing numbers of MZG translocate to it (*Gobius et al., 2016*). Immunohistochemistry for SOX9, a radial glia and astrocyte marker (*Sun et al., 2017*), revealed significantly more SOX9-positive MZG cell bodies at the pial surface of the IHF in the BTBR inbred strain at E14 and E15 compared to wildtype C57 mice (*Figure 7I–K* and *Supplementary file 1*). These MZG clustered along the IHF surface within 200 μm of the base of the IHF at E15

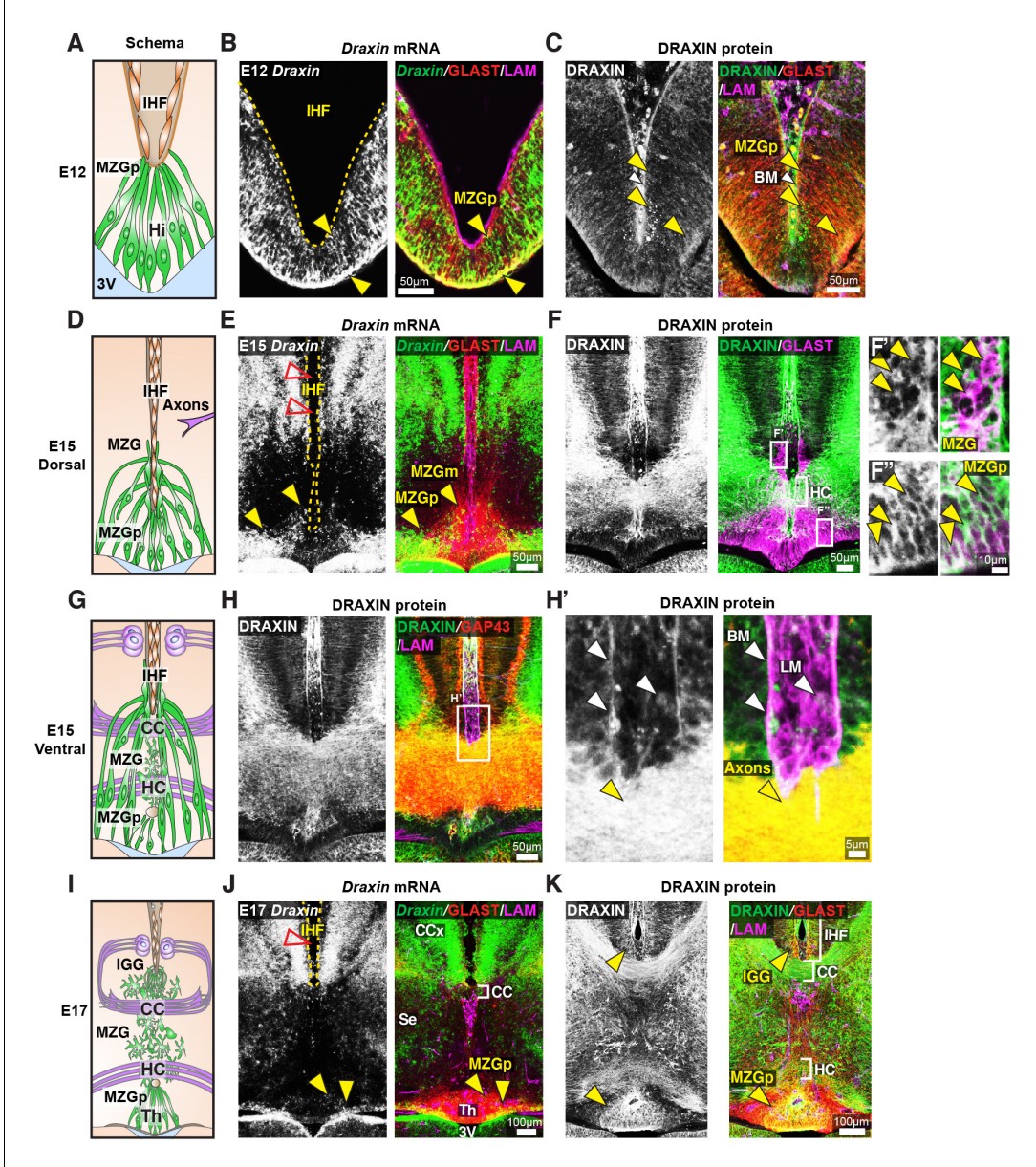

**Figure 6.** *Draxin* is expressed in midline zipper glia (MZG) and their progenitors, and associates with MZG membranes, leptomeninges, and the pial surface of the interhemispheric fissure (IHF). Schema of interhemispheric midline at E12 (**A**), E15 (**D** dorsal; **G** ventral) and E17 (**I**). In situ hybridisation for *Draxin* mRNA (white or green), with immunohistochemistry for astroglial marker, GLAST (red), and leptomeninges and IHF marker, Laminin (LAM; magenta) in E12 (**B**), E15 (**E**), and E17 (**J**) wildtype CD1 mid-horizontal telencephalic midline tissue sections. Yellow arrowheads indicate *Draxin*-positive/ GLAST-positive glia. Open red arrowheads indicate lack of *Draxin* mRNA within the IHF (yellow outlined). Immunohistochemistry for DRAXIN (white or green), GLAST (red or magenta), and LAM (magenta) in E12 (**C**), E15 (**F**), and E17 (**K**) wildtype CD1 mid-horizontal telencephalic midline tissue sections. (**H**) DRAXIN (white or green), axonal marker GAP43 (red), and LAM (magenta) in E15 ventral telencephalic midline tissue sections. Yellow arrowheads indicate regions of DRAXIN protein on GLAST-positive glial fibres (**C, F, K**) or DRAXIN protein on GAP43-positive axons (**H'**). White arrowheads indicate DRAXIN protein within the IHF and on the basement membrane of the IHF. BM: basement membrane; CCx: cingulate cortex; IGG: indusium griseum glia; LM: leptomeninges; MZGp: midline zipper glia progenitors; Se: septum; Th: telencephalic hinge; 3V: third ventricle.

(*Figure 7L* and *Supplementary file 1*). This did not result in narrowing the midline space between the adjacent hemispheres, indicating that earlier defects in the generation of the MZG population could underlie the persistence of the IHF in BTBR mice.

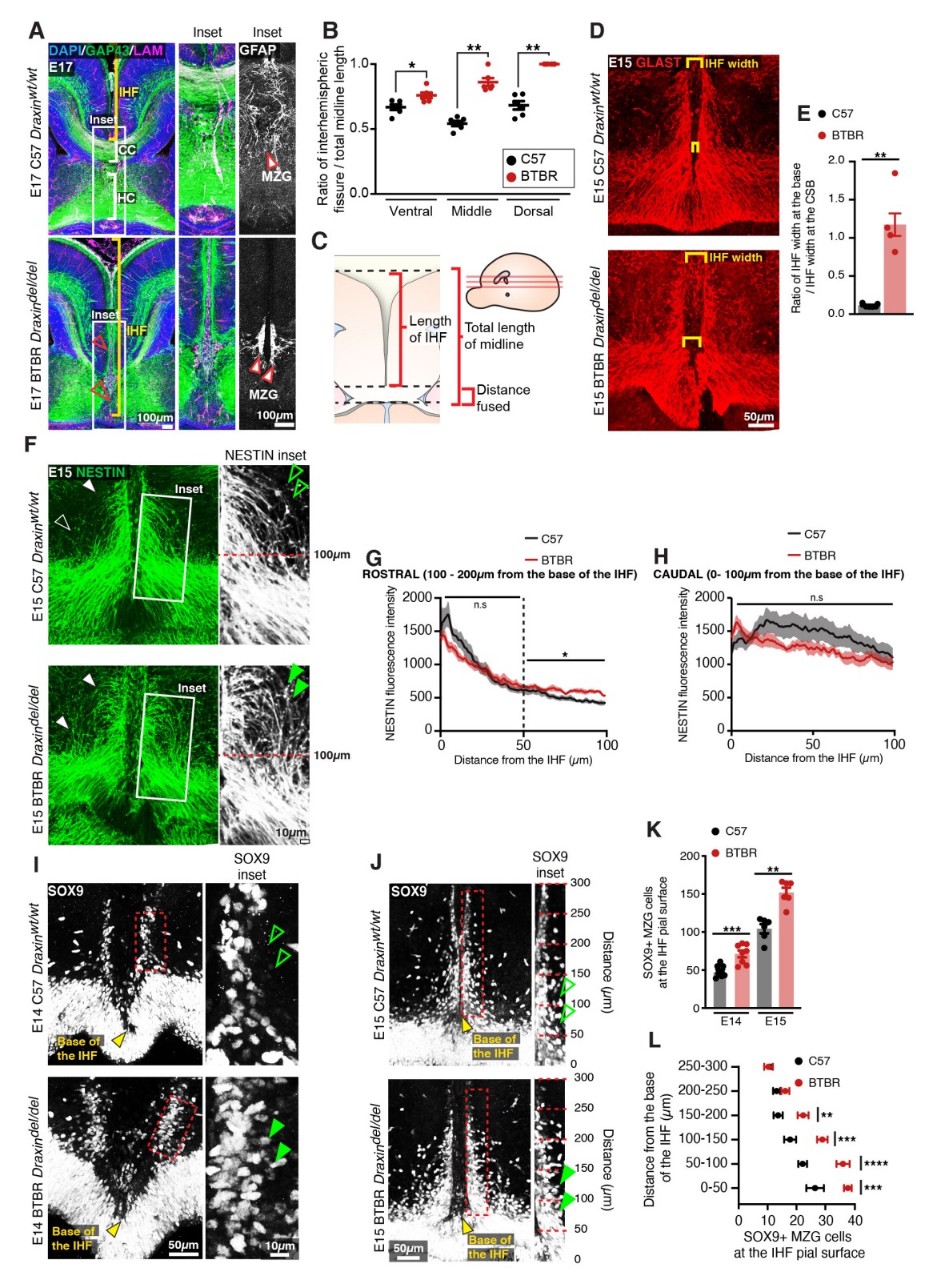

**Figure 7.** BTBR midline zipper glia (MZG) undergo precocious somal translocation to the interhemispheric fissure (IHF) and fail to intercalate for IHF remodelling. (**A**) Mid-horizontal sections of E17 wildtype C57 and BTBR mice immunolabelled with growing axon marker, GAP43 (green), astrocyte marker, GFAP (white; inset only), and leptomeninges marker, pan-LAMININ (magenta), and counterstained with DAPI (blue). The corpus callosum (CC) and hippocampal commissure (HC) are indicated with white brackets in C57 mice, and their absence in BTBR mice is indicated with red arrowheads. *Figure 7 continued on next page*

*Figure 7 continued*

The IHF is indicated with yellow brackets, and white boxes indicate the region where insets of GFAP-positive MZG were taken (right). (C) The ratio of IHF length over total telencephalic midline length was measured (B) from representative ventral, middle (shown in A), and dorsal horizontal sections. Immunohistochemistry on E15 wildtype C57 and BTBR horizontal brain sections labelling GLAST-positive MZG (D) and NESTIN-positive radial glia (F) at the ventral midline. IHF width is indicated with yellow brackets in (D), and the ratio of IHF length close to the base of the IHF compared with at the corticoseptal boundary (CSB) is quantified in (E). White arrowheads in (F) show radial MZG undergoing somal translocation to the IHF surface, and green arrowheads in insets demonstrate an increase in radial MZG fibres lateral to the base of the IHF in BTBR mice; the fluorescence intensity of these NESTIN fibres is quantified in (G) (rostral to the base of the IHF) and (H) (caudal, close to the base of the IHF). Immunohistochemistry on E14 (I) and E15 (J) wildtype C57 and BTBR horizontal brain sections labelling SOX9-positive glial cell bodies. Green arrowheads indicate increased SOX9-positive cell bodies at the IHF surface in BTBR mice. The base of the IHF is indicated with yellow arrowheads. The number of MZG cell bodies at the pial surface (outlined in red) is quantified in (K) and (L) (binned). Data represent mean ± SEM. *p<0.05, **p<0.01, ***p<0.001, ****p<0.0001 as determined with either an unpaired t test (E14, K), Mann–Whitney test (E15, K), or two-way ANOVA with Sidak's multiple comparison test (L).

The online version of this article includes the following source data for figure 7:

**Source data 1.** Ratio of interhemispheric fissure (IHF) over total midline length in E17 BTBR and C57 mice.

## Increased proliferation and cell cycle exit of MZG progenitors is associated with increased somal translocation of MZG and may underlie CC agenesis in BTBR mice

We have previously demonstrated increased proliferation of radial glia within the E14 cingulate cortex of the BTBR inbred mouse strain (*Faridar et al., 2014*). To determine whether a similar effect may underlie increased somal translocation of MZG in BTBR mice, we performed birth-dating of MZG progenitors by performing ethynyl deoxyuridine (EdU) injections and co-labelling with cell cycle marker, KI67. This analysis revealed an increase in MZG progenitors undergoing division in BTBR mice at E13 and E14, leading to significantly increased labelling of EdU-positive MZG at E14 and E15, respectively (*Figure 8A–D, F* and *Supplementary file 1*). Between E14 and E15, more dividing MZG progenitors exited the cell cycle (EdU-positive/KI67-negative; *Figure 8H* and *Supplementary file 1*) in BTBR mice compared with C57 controls. These differences were unlikely due to a mismatch in the developmental stage of the embryos since the length of the telencephalic midline was comparable between BTBR and wildtype C57 brains at the age of collection (*Figure 8E* and *Supplementary file 1*). We find that increased proliferation and cell cycle exit of MZG progenitors in BTBR mice leads to an over-abundance of NESTIN-positive MZG fibres and increased somal translocation of MZG to the IHF surface prior to IHF remodelling. Since we previously found that precocious generation of MZG is associated with disrupted IHF remodelling in mice with altered FGF8 signalling (*Gobius et al., 2016*), our results here suggest that disrupted IHF remodelling in BTBR mice could be due to precocious generation of MZG. Thus, loss of DRAXIN expression within MZG in BTBR mice disrupts the proliferation and migration of MZG to the IHF and consequent remodelling.

We further measured proliferation within the base of the IHF at the telencephalic hinge and found a significant increase in EdU-positive cells at E13 and E14, which were dividing at E12 and E13, respectively, in BTBR mice (*Figure 8A, B, I* and *Supplementary file 1*). The density of cells that remained in the cell cycle after dividing was significantly increased between E12 and E13 in BTBR mice (EdU-positive/KI67-positive; *Figure 8A , K* and *Supplementary file 1*). There were no differences in the density of cells that exited the cell cycle within the BTBR IHF compared to the wildtype C57 IHF (EdU-positive/KI67-negative; *Figure 8A–C, K* and *Supplementary file 1*). Cells within the IHF comprise CXCL12- and LAMININ-positive leptomeningeal cells, which are eliminated from the septum during IHF remodelling (*Gobius et al., 2016*). We conclude that failed IHF remodelling in BTBR mice is correlated with a transient period of *Hewitt, 1962* increased proliferation of both leptomeningeal cells within the IHF and MZG progenitors, and subsequent over-generation and migration of MZG to an enlarged IHF in BTBR mice.

## Discussion

The developmental basis of varied severity and expressivity of CCD in humans has to date been unclear. For example, humans with pathogenic variants in the DRAXIN receptor, *DCC*, display a range of CC phenotypes and associated HC malformations even across family members carrying the

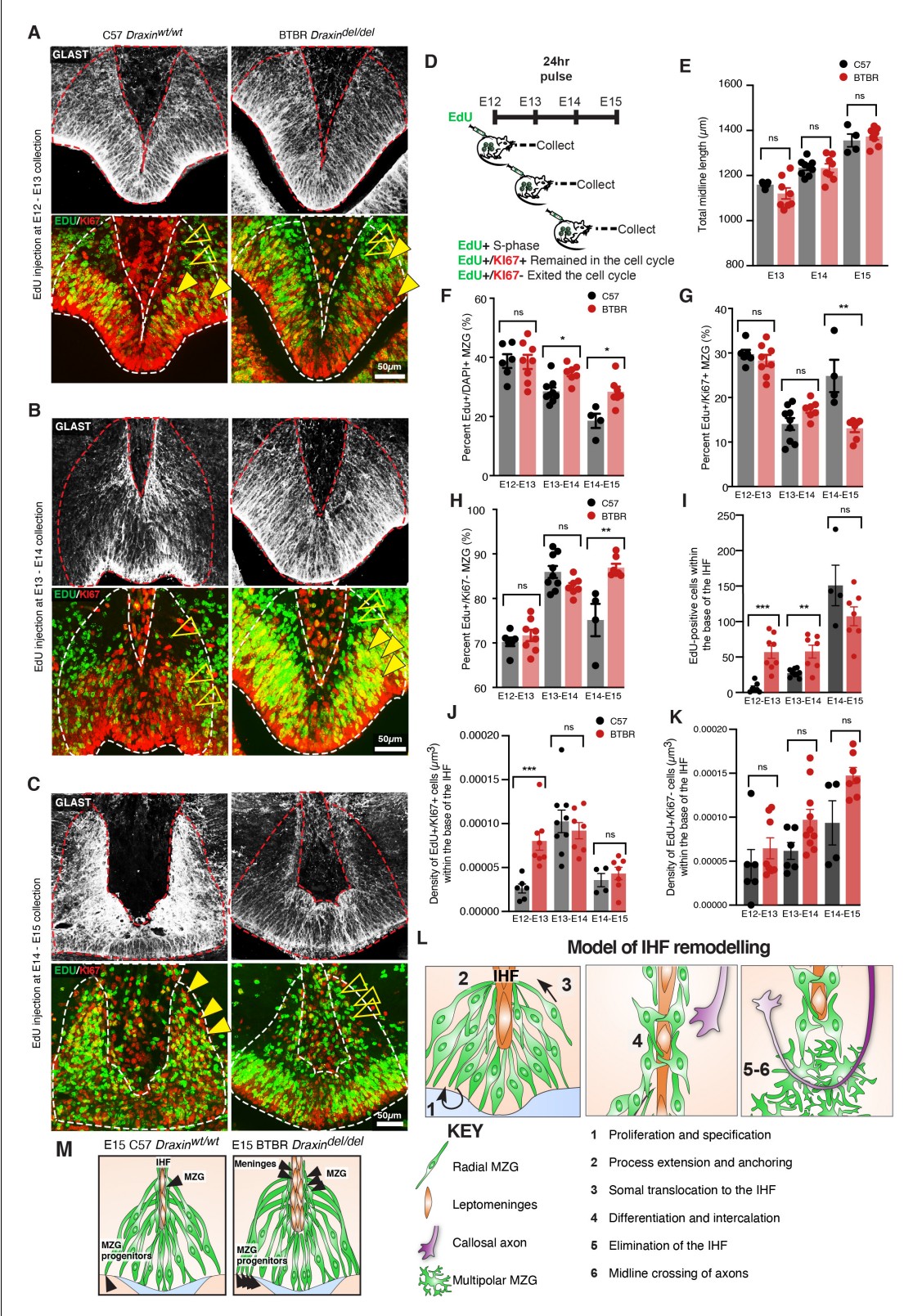

**Figure 8.** Elevated midline zipper glia (MZG) progenitor and leptomeningeal proliferation in BTBR mice. Wildtype C57 and BTBR pregnant mice were injected with ethynyl deoxyuridine (EdU) every 24 hr from E12 and collected 24 hr later (D). Immunohistochemistry for GLAST (white), EdU (green), and cell cycle marker, KI67 (red) on E13 (A), E14 (B), and E15 (C) wildtype C57 and BTBR horizontal brain sections of the telencephalic hinge and interhemispheric fissure (IHF) base. (E) To determine whether the litters were age-matched, the total length of the midline was compared between
*Figure 8 continued on next page*

Figure 8 continued

groups. The percentage of EdU-positive/DAPI-positive, EdU-positive/KI67-positive, and EdU-positive/KI67-negative MZG from the telencephalic hinge (white dotted outline) is quantified in (F), (G), and (H), respectively. EdU-positive cells within the base of the IHF is quantified in (I). EdU-positive/KI67-positive cells or EdU-positive/KI67-negative cells within the IHF were normalised to the total volume of the IHF as quantified in (J) and (K), respectively. Data represent mean ± SEM, *p<0.05, **p<0.01, ***p<0.001, ns = not significant, as determined with Mann–Whitney tests. (L) Schema of major steps involved in IHF remodelling. (M) Schema of BTBR phenotype at E15 compared with wildtype C57: BTBR mice display increased proliferation of MZG progenitors and precocious migration to the IHF surface as well as proliferation of the leptomeninges and expansion of the IHF, which may underlie failed IHF remodelling in these mice. See related *Figure 8—figure supplement 1*.

The online version of this article includes the following source data and figure supplement(s) for figure 8:

**Source data 1.** Measurements of IHF length and cells expressing EdU or KI67 within the telencephalic midline of E13-E15 BTBR and C57 mice.
**Figure supplement 1.** Volume of the interhemispheric fissure (IHF) base in BTBR mice.
**Figure supplement 1—source data 1.** Volume of the base of the IHF in E13-E15 BTBR and C57 mice.

same pathogenic variant (*Jamuar et al., 2017*; *Marsh et al., 2017*; *Marsh et al., 2018*; *Vosberg et al., 2019*). Here, we provide a comprehensive view of the genetic and cellular aetiology of CCD in a mouse model of variable CC and HC dysgenesis and demonstrate that a common structural aetiology might account for a wide range of human CCD phenotypes.

Our current and previously published (*Gobius et al., 2016*) findings indicate that the postnatal morphology of the CC in CCD is dependent on the available substrate formed by MZG-mediated IHF remodelling. IHF remodelling typically progresses in a ventral-to-dorsal and posterior-to-anterior manner, such that an earlier or more severe interruption of this process would be expected to result in a more severe reduction in available midline substrate, confined to the more ventral regions of the septum. It follows that later developing dorsal segments of the CC and HC will be more frequently disrupted than ventral segments, considering that the availability of a dorsal midline substrate for dorsal commissural axons depends on the earlier establishment of a permissive ventral substrate. Variable absence of a midline substrate has been previously associated with incomplete penetrance of CCD and HC dysgenesis in BALB/cWah1 and 129P1/ReJ inbred mouse strains (*Bohlen et al., 2012*; *Wahlsten et al., 2006*). Here, we find that the CC and dorsal HC are most commonly affected in BTBR and BTBR × C57 N2 mice, whereas the ventral HC, which spans the midline within the ventral septum in the mouse, is less affected (*Gobius et al., 2016*). Whilst the ventral HC is phylogenetically reduced or absent in humans (*Gloor et al., 1993*), we observed an analogous correlation between the length of IHF in partial CCD adults, which had not undergone remodelling during development, and the presence of a ventrally and anteriorly positioned CC remnant. Likewise, the posterior CC, which develops later than the anterior CC (*Hewitt, 1962*; *Rakic and Yakovlev, 1968*; *Ren et al., 2006*), was severely reduced in size in our human partial CCD cohort, with only one individual registering a near-normal CC width in this region. This suggests that milder perturbations of IHF remodelling in humans may still permit ventral crossing of callosal axons through a spatially restricted and ventrally positioned midline substrate, resulting in partial CCD. However, the progression of IHF remodelling during posterior CC development has not yet been characterised, so the extent to which these later-crossing axons are dependent on midline substrate formed by further MZG-mediated IHF remodelling remains to be investigated. Moreover, the anatomical position of callosal remnants in partial CCD can be diverse (*Tovar-Moll et al., 2007*; *Wahl et al., 2009*) and may not be fully represented in our cohort. Thus, a longitudinal study on IHF remodelling in normal human CC development and further investigation of IHF remodelling defects as a structural correlate for partial CCD with a variety of callosal remnants are needed to answer these questions. Nonetheless, our findings indicate that investigating the impact of candidate genetic causes for CCD on the competency of MZG to mediate IHF remodelling will potentially yield improvements in the precision of diagnosis and prognosis of CCD.

We identified an eight base-pair *Draxin* mutation as a predictor for severe CCD in BTBR mouse strains with incomplete penetrance. *Draxin* knockout mice display variable penetrance of CCD on a mixed or C57 genetic background (*Ahmed et al., 2011*; *Hossain et al., 2013*; *Islam et al., 2009*). The penetrance of CCD in our BTBR × C57 N2 mice carrying the *Draxin* mutation showed a similar penetrance to *Draxin* knockout mice on a C57 background suggesting that *Draxin* drives the CCD phenotype which is modified by additional genetic factors. A similar locus on chromosome 4 has previously been implicated in CC size of an intercross between NZB/BINJ and C57Bl/6By mice

(*Roy et al., 1998*). Therefore, mutations influencing *Draxin* function may underlie CCD in additional inbred mouse strains. While pathogenic variants in *DRAXIN* have not yet been reported in humans with CCD, humans with mutations resulting in haploinsufficiency in its receptor, *DCC*, display incomplete penetrance and variable expressivity of CCD (*Marsh et al., 2017*, *Marsh et al., 2018*, *Vosberg et al., 2019*). *Dcc* and *Draxin* interactions can determine the severity of CCD in mice (*Ahmed et al., 2011*). It is therefore possible that DRAXIN-DCC signalling also regulates IHF remodelling in humans through mechanisms similar to those elucidated here in mice.

DRAXIN has been classically described as an axon guidance ligand, which acts as a chemorepulsive cue for axons derived from cortical explants (*Islam et al., 2009*). Here, we show a distinct, earlier role for DRAXIN during CC formation in the astroglial-dependent formation of an interhemispheric substrate for axonal midline crossing. IHF remodelling is a multistep process involving (1) the generation and specification of MZG; (2) anchoring and extension of their (MZG) radial glial processes to both the third ventricle (apical) and to the IHF pial surface (basal); (3) MZG migration via somal translocation to the IHF; (4) MZG differentiation into multipolar astrocytes, including the elaboration of processes that penetrate the fissure; (5) the elimination of the leptomeninges within the IHF; and finally (6) the midline crossing of callosal and HC axons (*Figure 8L*). Most of these steps are disrupted in the absence of DRAXIN, but we propose that the greatest impact on the overall phenotype of BTBR mice comes from its regulation of the earliest steps of IHF remodelling (steps 1–3). More MZG are generated early in development which accumulate at the third ventricle and migrate a shorter distance along the midline pial surface (*Figure 8M*). These cells also have disorganised Nestin-positive radial processes, attached to an enlarged fissure that is filled with more leptomeningeal cells that increase their proliferation at early stages in the BTBR mouse (*Figure 8M*). Midline glial populations were previously reported to be abnormal in acallosal *Draxin* knockout mice (*Islam et al., 2009*), further supporting *Draxin* as the main genetic mediator of the BTBR midline phenotype. We previously demonstrated that dysfunction of these processes within MZG is causally associated with failed IHF remodelling in mice with altered expression of astrogliogenesis factors, FGF8, NFIA, and NFIB (*Gobius et al., 2016*). Whether DRAXIN acts downstream of FGF8-NFI signalling to regulate MZG development and IHF remodelling could be investigated in a future study. Moreover, DRAXIN could also regulate other aspects of MZG development, such as cell-cell and cell-extracellular matrix adhesion and signalling, or matrix metalloproteinase activity required for elimination of the leptomeninges.

In a companion study, we demonstrate that DCC and its chemoattractive ligand, NTN1, also regulate MZG morphology and migration to the IHF, and are therefore crucial for CC *Morcom et al., 2021* and HC formation in mice, as well as in humans with *DCC* mutations (*Morcom et al., 2021*). That study demonstrated that DCC and NTN1 promoted the extension of radial MZG fibres along the IHF and the migration of MZG to the IHF surface for remodelling. DRAXIN is a known antagonist of both NTN1 and DCC (*Ahmed et al., 2011*; *Gao et al., 2015*; *Islam et al., 2009*) and could therefore normally inhibit NTN1/DCC-mediated actin remodelling required for MZG morphology and somal translocation to the IHF. Under this model, loss of DRAXIN function in MZG would cause increased MZG somal translocation/migration, which we found to be the case in BTBR mice that do not express DRAXIN.

We found a significant increase in the proliferation of both MZG cells and leptomeningeal cells in BTBR mice homozygous for the *Draxin* mutation. This phenotype was not seen in DCC and NTN1 mutant mice and these molecules are not expressed or localised within the fissure (*Morcom et al., 2021*) whereas DRAXIN protein (but not mRNA) was observed on leptomeningeal cells. Together, this data suggests that DRAXIN may regulate cellular proliferation independent of NTN1/DCC signalling. DRAXIN is known to interact with the canonical WNT receptor LRP6 and antagonise canonical WNT signalling (*Miyake et al., 2009*). Moreover, canonical WNT signalling within cortical radial glia controls cell proliferation and astrogliogenesis (*Gan et al., 2014*). Thus, if LRP6 is also expressed by MZG or leptomeningeal cells, it may be the molecular link between DRAXIN and the regulation of cell proliferation and elevated generation of MZG and leptomeninges, which we observed in BTBR mice.

BTBR mice demonstrate an abnormally large IHF at the time of CC development that was directly associated with the increased proliferation of cells within the base of the IHF. DRAXIN is known to regulate the migration of cranial neural crest cells at earlier stages of development (*Hutchins and Bronner, 2018*) and is also involved in basement membrane remodelling during cranial neural crest

epithelial to mesenchyme transition in chicks (*Hutchins and Bronner, 2019*). Considering these roles, DRAXIN may directly regulate the leptomeninges, which are thought to arise from the neural crest (*Batarfi et al., 2017*), by controlling their proliferation or even repelling them from the site of IHF remodelling. Moreover, increased leptomeningeal cell-mediated expansion of the IHF in BTBR mice could prevent the intercalation of astroglial processes, impeding IHF remodelling further. Thus, DRAXIN plays multiple roles in CC formation, beginning with its role in MZG development, midline morphogenesis, and IHF remodelling before controlling callosal axon guidance (*Ahmed et al., 2011*; *Edwards et al., 2014*; *Islam et al., 2009*; *Morcom et al., 2015*). Whether the effect of the *Draxin* null-allele (*Islam et al., 2009*) and the BTBR-specific *Draxin* mutation on CC and HC formation is the same is yet to be determined.

Regarding the role of DRAXIN in partial versus complete CCD, our work suggests that the cumulative success of multiple processes under the control of DRAXIN may determine the extent of IHF remodelling. Moreover, the dysfunction of one or more of these processes associated with *Draxin* variants and co-inherited gene variants in signalling partners like DCC may instead determine the severity of the callosal phenotype. Conditional strategies that impact DRAXIN function in axons versus MZG versus leptomeninges and identification of further signalling partners in these cell populations are necessary to dissect these possibilities.

# Materials and methods

## Key resources table

| Reagent type (species) or resource | Designation | Source or reference | Identifiers | Additional information |
|---|---|---|---|---|
| Gene (*Mus musculus*) | *Draxin* | *Mus musculus* genome resource | 70433 | |
| Strain, strain background (*Mus musculus*) | BTBR T + Itpr3tf/J | The Jackson Laboratory | 002282 | |
| Strain, strain background (*Mus musculus*) | C57Bl/6J | The Jackson Laboratory | 000664 | |
| Cell line (*Homo sapiens*) | Human embryonic kidney (HEK) 293T | ATCC | RRID:CVCL_0045 | ATCC Cat# CRL-1573, obtained via the University of Queensland |
| Antibody | Sheep polyclonal anti-DRAXIN | R&D Systems | AF6149, RRID:AB_10640005 | '(1:250)' |
| Antibody | Mouse monoclonal anti-human KI67 | BD Pharmingen | 550609, RRID:AB_393778 | '(1:500)' |
| Antibody | Mouse monoclonal anti-GAP43 | Millipore | MAB347, RRID:AB_94881 | '(1:500)' |
| Antibody | Rabbit polyclonal anti-GFP | Thermo Fisher Scientific | A-6455, RRID:AB_221570 | '(1:1000)' |
| Antibody | Rabbit polyclonal anti-GFAP | Dako | Z0334, RRID:AB_10013382 | '(1:500)' |
| Antibody | Mouse monoclonal anti-Glast (EAAT1) | Abcam | Ab49643, RRID:AB_869830 | '(1:500)' |
| Antibody | Rabbit polyclonal anti-Glast (EAAT1) | Abcam | Ab416, RRID:AB_304334 | '(1:250)' |
| Antibody | Chicken polyclonal anti-Laminin | LS-Bio | C96142, RRID:AB_2033342 | '(1:500)' |
| Antibody | Rabbit polyclonal anti-Laminin (pan-Laminin) | Sigma | L9393, RRID:AB_477163 | '(1:500)' |

*Continued on next page*

*Continued*

| Reagent type (species) or resource | Designation | Source or reference | Identifiers | Additional information |
|---|---|---|---|---|
| Antibody | Rat monoclonal anti-Nestin (NES) | Developmental Studies Hybridoma Bank | AB 2235915, RRID:AB_2235915 | '(1:50)' |
| Antibody | Rabbit polyclonal anti-SOX9 | Merck | AB5535, RRID:AB_2239761 | '(1:500)' |
| Antibody | Goat anti-β-ACTIN | SCIGEN | AB0145-200, N/A | '(1:1000)' |
| Recombinant DNA reagent | pPBCAG-IRES-GFP | *Chen et al., 2020* | | |
| Recombinant DNA reagent | pGEMT | Promega | A1360 | |
| Sequence-based reagent | *Draxin_*riboprobe_F | Allen Brain Atlas | | CAGGGAGGTTTAGGACAAACAG |
| Sequence-based reagent | *Draxin_*riboprobe_R | Allen Brain Atlas | | TGTAGGAGCTGAGGGAAAGAAG |
| Sequence-based reagent | *Draxin_*CDS_F | This paper | | GAATTCGACAGGGAGAGCCAATG |
| Sequence-based reagent | *Draxin_*CDS_R | This paper | | GCGGCCGCGTACTGGGCGTACACCTGCT |
| Sequence-based reagent | Chromosome 4, SNP rs6397070 forward | This paper | | TTTATGGCTGGGGACTTCAG |
| Sequence-based reagent | Chromosome 4, SNP rs6397070 reverse | This paper | | CGAATCCAAAGCTCTCTTGC |
| Sequence-based reagent | Chromosome 9, SNP rs29890894, forward | This paper | | AGCTTGGTGGCATCCATATC |
| Sequence-based reagent | Chromosome 9, SNP rs29890894, reverse | This paper | | GCACTCTCCCTACTGCTTGG |
| Sequence-based reagent | Chromosome 15, SNP rs31781085 forward | This paper | | GATCGTTGCAGTGACCACAC |
| Sequence-based reagent | Chromosome 15, SNP rs31781085 reverse | This paper | | GCTGATTGGCAGGTTCTGAT |
| Sequence-based reagent | *Draxin* allele genotyping, wildtype forward | This paper | | AGACGGTCCCTGCGTCTC |
| Sequence-based reagent | *Draxin* allele genotyping, mutant forward | This paper | | GTCGCAGACGGTCCCTTG |
| Sequence-based reagent | *Draxin* allele genotyping, common reverse | This paper | | AGGCTTCCCAGATGACACTC |
| Commercial assay, kit | Click-iT EdU Cell Proliferation Kit for Imaging, Alexa Fluor 488 dye | Invitrogen | C10337 | |
| Commercial assay, kit | Click-iT EdU Cell Proliferation Kit for Imaging, Alexa Fluor 555 dye | Invitrogen | C10338 | |
| Software, algorithm | Fiji | Fiji | RRID:SCR_002285 | |
| Software, algorithm | Prism | GraphPad | RRID:SCR_002798 | |

*Continued on next page*

*Continued*

| Reagent type (species) or resource | Designation | Source or reference | Identifiers | Additional information |
|---|---|---|---|---|
| Software, algorithm | Imaris | Bitplane | N/A | |
| Other | 4′,6-Diamidino-2-phenylindole dihydrochloride (DAPI) | Invitrogen | D1306 | '(1:750)' |

## Animals

BTBR, CD1, and C57 mice were bred and tested at The University of Queensland according to the Australian Code of Practice for the Care and Use of Animals for Scientific Purposes and with prior ethics approval from The University of Queensland Animal Ethics Committee. To generate BTBR × C57 N2 mice with varying degrees of CCD, an N1 intercross was produced and N1 females were crossed with BTBR males to generate N2 mice for analysis. The day of birth was designated postnatal day (P)0. Time-mated females were obtained by housing male and female mice together overnight and the following morning was designated embryonic day (E)0. Animals were anaesthetised and collected as previously described (*Suárez et al., 2014*). For EdU labelling, pregnant dams were given an intraperitoneal injection of 5-ethynyl-2′-deoxyuridine (EdU; 5 mg per kg body weight at E14 or 7.5 mg per kg body weight at E12 and E13, Invitrogen) and embryos were collected 24 hr later. Sex was not determined for embryonic studies. A total of 112 adult BTBR × C57 N2 mice were analysed (n = 47 males, n = 65 females). For the analysis of adult mouse tissue, mice were perfused at 12 weeks of age.

Genotyping of BTBR × C57 N2 mice was performed by PCR of SNPs associated with CC size in the BTBR × C57 F2 intercross as previously described by us (*Jones-Davis et al., 2013*). Amplicons were generated for SNP regions using the following primers: chromosome 4, SNP rs6397070, forward TTTATGGCTGGGGACTTCAG and reverse CGAATCCAAAGCTCTCTTGC; chromosome 9, SNP rs29890894, forward AGCTTGGTGGCATCCATATC, reverse GCACTCTCCCTACTGCTTGG; chromosome 15, SNP rs31781085, forward GATCGTTGCAGTGACCACAC, reverse GCTGA TTGGCAGGTTCTGAT. Genotyping for the *Draxin* mutation was performed by PCR using the following primers: wildtype forward AGACGGTCCCTGCGTCTC, mutant forward GTCGCAGACGGTCCC TTG, and common reverse AGGCTTCCCAGATGACACTC. Sanger sequencing was performed at the Australian Genome Research Facility.

## Human participants

Participants were enrolled in an ongoing study at The University of Queensland from October 2014 with approval from The University of Queensland Research Ethics committee. 10 individuals with partial CCD aged 21–72 (mean age = 40 years, standard deviation = 17.73) and 9 neurotypical individuals (mean age = 34.89, standard deviation = 16.77) provided written informed consent. MRI scans of all participants were reviewed by a neuroradiologist (SM), who has extensive experience in human brain malformations. The groups did not differ significantly in either age or gender. Genetic information to determine cause of CCD was not collected for this study.

## MRI scans

MRI scanning was performed using the same parameters and cohort of BTBR × C57 N2 adult mice as described in *Edwards et al., 2020*. Briefly, skulls were post-fixed in 4% paraformaldehyde for at least 48 hr after perfusion. Following this, each BTBR × C57 N2 adult mouse skull (with brain in situ) was washed in phosphate buffered saline (PBS) with 0.2% sodium azide. Skulls were immersed in Fomblin Y-LVAC fluid (Solvay Solexis, Bollate, Italy), and air was actively removed from each sample via vacuum pumping prior to MRI scanning. All skulls (n = 112) initially underwent two-dimensional scans measuring diffusion in the mediolateral direction to identify the commissural phenotype. From this analysis, a subset of brains of each of complete CCD (n = 10), partial CCD (n = 11), and full CC (n = 10) phenotypes were dissected from skulls, incubated for 4 days in 0.2% gadopentetate dimeglumine (Magnevist, Berlex Imaging, Wayne, NJ, USA), and scanned with a 16.4 Tesla Bruker Avance MRI scanner using a FLASH sequence that was described previously (*Schanze et al., 2018*): FLASH sequence (voxel size = 0.03 × 0.03 × 0.03 mm, MTX 654 × 380 × 280, FOV 19.6 × 11.4 × 8.4 mm,

TR = 50 ms, TE = 12 ms, flip angle of 30°). Three adult BTBR inbred mouse brains were scanned using identical processing steps. C57 mouse brain scans were acquired previously for *Ullmann et al., 2013* and were kindly provided by Dr. Nyoman Kurniawan (Centre for Advanced Imaging, The University of Queensland, Australia).

Human participants underwent MRI at the Centre for Advanced Imaging (The University of Queensland) using a 7 Tesla Siemens Magnetom whole-body MRI scanner. Structural MRI data was acquired as described previously (*Hearne et al., 2019*): at 7 T, MP2RAGE sequence (voxel size = 0.75 × 0.75 × 0.75 mm, MTX 256 × 300 × 320, FOV 192 × 225 × 240 mm, TR = 4300 ms, TE = 3.44 ms, TI = 840/2370 ms, flip angle of 5°, FOV 192 × 225 × 240 mm).

## MRI anatomical measurements

Commissure and brain sizes were measured in OsiriX (v 5.8.5; *Rosset et al., 2004*), blind to animal genotype. All commissure lengths and areas were measured in the midsagittal plane on single-direction diffusion scans. Anteroposterior CC length was measured using the straight length tool from the most anterior point to the most posterior point of the CC. HC length was measured using the straight length tool from the dorsal-most aspect of the HC (inferior to the CC) to the ventral-most aspect of the HC. HC area was measured using the closed polygon tool using the same superior and inferior boundaries for HC length. Anterior commissure area was measured using the closed polygon tool. Brain length was measured using the straight length tool, from the anterior aspect of the frontal pole to the posterior-most aspect of the cerebellum. IHF and CC measurements from human MRI were measured from representative images displayed within figures using ITK-SNAP v3.8.0 (*Yushkevich et al., 2006*).

## Immunohistochemistry and tissue staining

Immunohistochemistry was performed on 50 µm tissue sections as previously described (*Moldrich et al., 2010*). Primary antibodies used: sheep anti-DRAXIN (1:250; AF6149, R&D Systems), mouse anti-human KI67 (1:500; 550609, BD Pharmingen), mouse anti-GAP43 (1:500; MAB347, Millipore), rabbit anti-GFAP (1:500; Z0334, Dako), mouse anti-GLAST (or EAAT1; 1:500; ab49643, Abcam), rabbit anti-GLAST (or EAAT1; 1:250; ab416, Abcam), chicken anti-LAMININ (1:250; LS-C96142, LSBio), rabbit anti-LAMININ (1:250; L9393, Sigma), rat anti-NESTIN (AB 2235915, DSHB), and rabbit anti-SOX9 (1:500, AB553, Merck). Secondary antibodies were Alexa Fluor IgG antibodies (1:500, Invitrogen) or biotinylated IgG antibodies (1:500 or 1:1000, Jackson Laboratories) used in conjunction with Alexa Fluor 647-conjugated streptavidin (1:500, Invitrogen) amplification. EdU labelling was performed using the Click-iT EdU Alexa Fluor 488 or Alexa Fluor 555 Imaging kits (Invitrogen) according to the manufacturer's instructions. Cell nuclei were labelled using 4′,6-diamidino-2-phenylindole dihydrochloride (DAPI, Invitrogen) and coverslipped using ProLong Gold anti-fade reagent (Invitrogen) as mounting media.

## In situ hybridisation

In situ hybridisation was performed as previously described (*Moldrich et al., 2010*), with the following minor modifications: Fast red (Roche) was applied to detect probes with fluorescence. For fluorescent in situ hybridisation against *Draxin* mRNA in wildtype CD1 mice, the *Draxin* CDS was amplified by PCR using the following primer pairs from the Allen Brain Atlas (*Lein et al., 2007*): CAGGGAGGTTTAGGACAAACAG and TGTAGGAGCTGAGGGAAAGAAG. The *Draxin* CDS was subsequently cloned into the pGEM-T Vector (Promega USA) and sequences were verified. Digoxygenin-labelled (DIG RNA labelling mix; Roche) antisense riboprobes were also generated in a similar manner from the *Draxin* CDS amplified from BTBR and C57 E15 telencephalic midline tissue using the following primer pairs: CGACAGGGAGAGCCAATG and GTACTGGGCGTACACCTGCT.

## Generation of BTBR and C57 *Draxin* expression plasmids

The *Draxin* CDS was amplified from BTBR and C57 E15 telencephalic midline tissue using the following primer pairs containing added EcoRI and NotI restriction sites: GAATTCGACAGGGAGAGCCAATG and GCGGCCGCGTACTGGGCGTACACCTGCT. The amplified sequence was digested with EcoRI and NotI and subsequently inserted into the pPBCAG-IRES-GFP (pPBCAGIG) expression

plasmid (*Chen et al., 2020*) via the corresponding restriction sites. Successful cloning of the *Draxin* CDS into the pPBCAGIG plasmid was verified by Sanger sequencing.

## Western blot

Whole-cell protein extracts were prepared from transfected HEK293T cells and dissected midline tissue as described previously (*Bunt et al., 2010*). Protein extracts were cleared by centrifugation and used for western blotting as described previously (*Bunt et al., 2017*). Primary antibodies used for immunoblotting were sheep anti-DRAXIN (AF6149, R&D Systems, 1 µg/mL), goat anti-β-ACTIN (AB0145-200, SICGEN, 1:1000 or 3 µg/mL), and rabbit anti-GFP (A-6455, Thermo Fisher Scientific, 1:1000). The secondary antibodies used were sheep IgG (H and L) antibody DyLight 800 conjugated (Rockland Immunochemicals Inc, 1:15000), IRDye 680LT donkey anti-rabbit (LI-COR, 1:15000), and IRDye 800CW donkey anti-goat (LI-COR, 1:15000). Immunoblotted membranes were imaged using the Odyssey Classic (LI-COR) and Image Studio 5 software (LI-COR).

## Image acquisition and analysis

Microscopy for fluorescence immunohistochemistry or in situ hybridisation was performed using either an inverted Zeiss Axio-Observer fitted with a W1 Yokogawa spinning disk module, Hamamatsu Flash4.0 sCMOS camera, and Slidebook 6 software or an inverted Nikon TiE fitted with a Spectral Applied Research Diskovery spinning disk module, Hamamatsu Flash4.0 sCMOS camera, and Nikon NIS software. Pseudocoloured image projections of ~10–20 µm thick z-stacks were acquired. Images of chromogenic in situ hybridisation samples were acquired on a Zeiss upright Axio-Imager Z1 microscope with Axio- Cam HRm camera and Zen software (Carl Zeiss). Images were cropped, sized, and contrast-brightness enhanced for presentation with ImageJ and Photoshop software (Adobe).

The ratio of IHF length to total telencephalic midline length was quantified in ImageJ freeware (National Institutes of Health, Bethesda, USA) as previously described (*Gobius et al., 2016*). The IHF width was measured from LAMININ and GLAST stained tissue sections in two different regions: (1) ~5 µm from the base of the IHF and (2) at the most rostral region where GLAST-positive MZG fibres attach to the IHF surface which coincided with the corticoseptal boundary. The IHF width at the base (region 1) was then expressed as a ratio over the IHF width at the corticoseptal boundary (region 2) in order to reflect changes in the compression of the IHF at the base prior to IHF remodelling. Fluorescence intensity of GLAST-positive and NESTIN-positive MZG fibres in a region of interest ~100 × 200 µm (medial-lateral × rostral-caudal) along the IHF surface was measured in ImageJ v1.52i freeware from multiple intensity projection 2D images generated from 3D z-stacks. The number of SOX9-positive MZG cell bodies was counted manually using the Cell Counter plugin in ImageJ v1.51s freeware from a region of interest at the surface of the IHF measuring ~10 × 200 µm (medial-lateral × rostral-caudal). For cell proliferation assays, the telencephalic hinge was outlined as a region of interest in Imaris from 3D z-stacks. The number of DAPI-positive, EdU-positive, KI67-positive cells was automatically counted using the Imaris spots function as previously described (*Faridar et al., 2014*). The colocalisation function was used to determine co-labelled spots within 4 µm (MZG) or 5 µm (leptomeninges) of each other. All counts were performed blind to the experimental conditions and normalised to the number of DAPI-positive cells or the volume of the region of interest.

## Experimental design and statistical analysis

Statistical tests were performed in GraphPad Prism (v.7 and v.8), and $p < 0.05$ was considered significant. Statistical testing was performed for all quantitative analyses – the relevant statistical test performed, number of biological replicates, and level of significance are described for all quantitative results in figures or their legends, and in *Supplementary file 1*. Data sets were first checked for normality using a D'Agostino–Pearson omnibus normality test. If the data was not normally distributed, then the non-parametric equivalent test was performed. For qualitative comparisons (*Figures 4* and *8*), a minimum of 3 animals were used for each experiment.

## Acknowledgements

We thank Aiman Al Najjar, Nicole Atcheson, and Dr. Nyoman Kurniawan for assistance in conducting MRI at the Centre for Advanced Imaging, The University of Queensland. We thank Rumelo Amor, Arnaud Guardin, and Andrew Thompson for assistance with microscopy, which was performed in the Queensland Brain Institute's Advanced Microscopy Facility. We thank the staff of The University of Queensland Biological Resources for care and breeding of animals. This work was supported by Australian National Health and Medical Research Council (NHMRC) grants GNT1048849 and GNT1126153 to LJR, Australian Research Council (ARC) grant DP200102363 to LJR and US National Institutes of Health grant 5R01NS058721 to EHS and LJR. RS received an ARC DECRA fellowship (DE160101394). LM and JWCL were supported by a Research training program scholarship (Australian Postgraduate Award). TJE and KSC were supported by a University of Queensland Research Scholarship. LM, TJE, and JWCL also received Queensland Brain Institute Top-Up scholarships. RJD was supported by Brain Injured Childrens After-Care Recovery Endeavours (BICARE) Inc LJR was supported by an NHMRC Principal Research Fellowship (GNT1120615).

We thank the families and members of the Australian Disorders of the Corpus Callosum (Aus-DoCC) for their support and time in being involved in this research.

## Additional information

### Funding

| Funder | Grant reference number | Author |
| --- | --- | --- |
| National Health and Medical Research Council | GNT1048849 | Linda J Richards |
| National Health and Medical Research Council | GNT1126153 | Linda J Richards |
| National Institutes of Health | 5R01NS058721 | Elliott H Sherr<br>Linda J Richards |
| Australian Research Council | DE160101394 | Rodrigo Suárez |
| Department of Education, Employment and Workplace Relations, Australian Government | Research Training Program scholarship | Laura Morcom<br>Jonathan WC Lim |
| University of Queensland | Research Scholarship | Timothy J Edwards<br>Kok-Siong Chen |
| Queensland Brain Institute | Top-Up Scholarship | Laura Morcom<br>Timothy J Edwards<br>Jonathan WC Lim |
| National Health and Medical Research Council | GNT1120615 | Linda J Richards |
| Brain Injured Childrens After-Care Recovery Endeavours (BICARE) Inc | | Ryan J Dean |
| Australian Research Council | DP200102363 | Linda J Richards |

The funders had no role in study design, data collection and interpretation, or the decision to submit the work for publication.

### Author contributions

Laura Morcom, Timothy J Edwards, Conceptualization, Data curation, Formal analysis, Investigation, Visualization, Methodology, Writing - original draft, Writing - review and editing; Eric Rider, Jonathan WC Lim, Investigation, Methodology, Writing - review and editing; Dorothy Jones-Davis, Resources, Investigation, Methodology, Writing - review and editing; Kok-Siong Chen, Investigation, Writing - review and editing; Ryan J Dean, Data curation, Investigation, Writing - review and editing; Jens Bunt, Ilan Gobius, Rodrigo Suárez, Conceptualization, Supervision, Investigation, Writing - review and editing; Yunan Ye, Formal analysis, Investigation, Writing - review and editing; Simone

Mandelstam, Formal analysis, Validation, Investigation, Methodology, Writing - review and editing; Elliott H Sherr, Conceptualization, Formal analysis, Supervision, Funding acquisition, Project administration, Writing - review and editing; Linda J Richards, Conceptualization, Resources, Supervision, Funding acquisition, Investigation, Writing - original draft, Project administration, Writing - review and editing

### Author ORCIDs
Laura Morcom (iD) https://orcid.org/0000-0001-6683-4356
Timothy J Edwards (iD) https://orcid.org/0000-0002-2175-7964
Eric Rider (iD) https://orcid.org/0000-0002-4019-0238
Jonathan WC Lim (iD) http://orcid.org/0000-0002-5074-6359
Kok-Siong Chen (iD) https://orcid.org/0000-0001-5796-5290
Ryan J Dean (iD) https://orcid.org/0000-0002-7146-1352
Jens Bunt (iD) https://orcid.org/0000-0003-0397-2019
Yunan Ye (iD) https://orcid.org/0000-0001-9084-6314
Ilan Gobius (iD) https://orcid.org/0000-0003-3255-2531
Rodrigo Suárez (iD) https://orcid.org/0000-0001-5153-5652
Elliott H Sherr (iD) https://orcid.org/0000-0002-4118-5385
Linda J Richards (iD) https://orcid.org/0000-0002-7590-7390

### Ethics
Human subjects: Ethics for human experimentation was acquired by local ethics committees at The University of Queensland (Australia), and carried out in accordance with the provisions contained in the National Statement on Ethical Conduct in Human Research and with the regulations governing experimentation on humans (Australia), under the following human ethics approvals: HEU 2014000535, and HEU 2015001306.

Animal experimentation: Prior approval for all breeding and experiments was obtained from the University of Queensland Animal Ethics Committee and was conducted in accordance with the Australian code for the care and use of animals for scientific purposes. The protocol, experiments and animal numbers were approved under the following project approval numbers: QBI/305/17, QBI/306/17, QBI/311/14 NHMRC (NF), QBI/356/17, and QBI/310/14/UQ (NF).

### Decision letter and Author response
Decision letter https://doi.org/10.7554/eLife.61618.sa1
Author response https://doi.org/10.7554/eLife.61618.sa2

## Additional files
### Supplementary files
- Supplementary file 1. Statistics.
- Transparent reporting form

### Data availability
All data generated or analysed during this study are included in the manuscript and supporting files. Source data files have been provided for all figures that contain numerical data.

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
