## [Decision Letter]

**Acceptance summary:**

Your manuscript is a comprehensive study of the cellular and genetic mechanisms involved in the diversity of corpus callosum dysgenesis (CCD) phenotypes and associated hippocampal commissure (HC) malformations in humans with pathogenic variants in the DRAXIN receptor, DCC , in comparison to a mouse model with this mutation. This study extends your work over the years revealing that interhemispheric fissure (IHF) fusion is critical for proper formation of the corpus callosum and its failure is the main cause of complete CCD, and that the extent of aberrant IHF remodeling correlates with commissure dysgenesis severity. The range of CC phenotypes and associated HC malformations in the mouse mirror those in humans and are also variable, perhaps related to stochasticity on the mechanisms involved, or to the dependency on other allelic variants. In all, this is a fine study.

**Decision letter after peer review:**

Thank you for submitting your article "DRAXIN regulates interhemispheric fissure remodelling to influence the extent of corpus callosum formation" for consideration by *eLife*. Your article has been reviewed by 3 peer reviewers, and the evaluation has been overseen by a Reviewing Editor and Huda Zoghbi as the Senior Editor. The following individuals involved in review of your submission have agreed to reveal their identity: Cecilia Flores (Reviewer #1); Thomas Pratt (Reviewer #2).

The reviewers have discussed the reviews with one another and the Reviewing Editor has drafted this decision to help you prepare a revised submission.

Summary:

Your manuscript is an excellent account of the cellular and genetic mechanisms involved in the diversity of corpus callosum dysgenesis (CCD) phenotypes in humans and in a mouse model. Your work over the years has revealed that interhemispheric fissure (IHF) fusion is critical for proper formation of the callosum and its failure is the main cause of complete CCD. Here you nicely show that the extent of aberrant interhemispheric fissure (IHF) remodeling does in fact correlate with commissure dysgenesis severity, in inbred and outcrossed BTBR mouse strains, as well as in humans with partial CCD. The phenotypes in the mouse are very similar to what is found in humans, and also variable, perhaps related to stochasticity on the mechanisms involved, or to the dependency on other allelic variants.

You also identify an eight base pair deletion in Draxin and misregulated astroglial and leptomeningeal proliferation as genetic and cellular factors for variable IHF remodelling and CCD in BTBR acallosal strains. The Draxin mutations interrupt the normal remodeling (closing) of interhemispheric fissure necessary for callosal axons to cross. Your study thus places the focus on midline cellular populations and away from axonal navigation as the main source of corpus callosum dysgenesis. The findings are important to understand what mutations cause CCD in humans and how, mechanistically, it occurs.

The reviewers agreed that the data and experiments as they stand are sufficient for publication in *eLife*. Suggestions for amendments which were indicated both in the reviews and a Consultation Session with the reviewers, and include:

1. Draxin mutations interrupt the normal remodeling (closing) of interhemispheric fissure necessary for callosal axons to cross. One question is how, cellularly or molecularly, Draxin implements glial and leptomeningeal cells remodel the interhemispheric fissure. Proliferation by itself as shown in Figure 8 does not seem to explain the phenotypes. The model that you are proposing is not fully clear. Does Draxin affect cell-cell adhesion, cell-cell signaling, membrane processes, metaloproteinase activity? You could allude to these possibilities in the discussion.

2. Reviewer 2 asks that the human data be more clearly integrated into the rest of the study. The phenotypes in the mouse are very similar to what is found in humans, and also variable, perhaps related to stochasticity on the mechanisms involved, or to the dependency on other allelic variants. MRI scans of human subjects with a spectrum of CC abnormalities, in addition, shows that commissure abnormalities correlate with midline fusion defects. The human data rely on adult phenotypes and do not relate to Draxin and although it is interesting in itself it gives little insight into the embryonic mechanisms that are so nicely explored in the mouse part of the study. In my opinion, the data should be included but the text in the Results and Discussion to better blend these two aspects.

3. The reviewers thought that the use of BTBR x C57 N2 crosses where commissure phenotype is correlated with the Draxin mutation (Figure 5) is a nice illustration of unpicking variable penetrance. However, the description of the findings that prompted you to investigate the role of Draxin in CCD needs to be clearer.

Also, It seems that the Draxin deletion does not affect HC formation. However, in the Results section you state that "To investigate how DRAXIN regulates CC and HC formation…". This is confusing. Also, it appears that the effect varies between BTRB mice and the BTRB x C57 cross, but this is not discussed clearly. One solution is to move the Draxin findings to the next to the lat last part of the Results, before the human results.

4. As Reviewing Editor: You say that contrary to published evidence on Draxin as a known regulator of axon guidance, "Draxin was not found to regulate axon guidance in this context, but rather impacted the proliferation of astroglia and leptomeninges; two cell populations that are also not yet considered to play a major role in formation of commissures". And "while pathogenic variants in DRAXIN have not yet been reported in human individuals with CCD, a similar effect may underlie the spectrum of phenotypes observed in humans with DCC mutations, since Dcc and Draxin have been demonstrated to interact to determine the severity of CCD in mice (Ahmed et al., 2011)". In the Introduction and/or in the Discussion it would be welcome to mention the relationship of Draxin to DCC as receptors for Netrin , as well as your other submitted study that is on BioRxiv and submitted to *eLife* on Dcc and Netrin1 regulating intrahemispheric fusion and the midline zipper glia.

5. Statistics:

a. For Figure 1, there is very little information about statistical analysis. For figure 1 C, it needs to be explained why the Welsh test was used instead of a one-way ANOVA. The errors on the bars do not seem to correspond to SEM; this needs to be clarified.

b. For Figures 3 G and H, if the data are presented in single graphs, it is not clear why unpaired t tests or Mann-Whitney tests were conducted (instead of ANOVAs). Why a non-parametric test was used is not explained.

c. Some of the data are not normally distributed (particularly clear for pink data points in Figure 5a,e,i,m) so it is not appropriate to show standard errors (the SEM bars could simply be removed), a non-parametic Kruskal-Wallis ANOVA has been used which is appropriate.

Revisions expected in follow-up work:

1. You could perform additional experiments on the glial cells in which Draxin is expressed, to better understand mechanism.

2. A central contention of this study is that variable penetrance of the commissure phenotypes in the BTBR x C57 mice stems from an earlier midline fusion phenotype. It would have been useful to discern whether the (embryonic) midline fusion phenotype also showed the same partial penetrance in BTBR x C57 mice, perhaps also correlated with the WT/MUT Draxin alleles (as in Figure 5). This would be a testable prediction of the hypothesis that midline fusion mediates the Draxin phenotype.

---

## [Author Response]

Revisions for this paper:1. Draxin mutations interrupt the normal remodeling (closing) of interhemispheric fissure necessary for callosal axons to cross. One question is how, cellularly or molecularly, Draxin implements glial and leptomeningeal cells remodel the interhemispheric fissure. Proliferation by itself as shown in Figure 8 does not seem to explain the phenotypes. The model that you are proposing is not fully clear. Does Draxin affect cell-cell adhesion, cell-cell signaling, membrane processes, metaloproteinase activity? You could allude to these possibilities in the discussion.

Our current model is that IHF remodelling is a multi-step process involving (1) the generation and specification of MZG; (2) anchoring and extension of their (MZG) radial glial processes to the third ventricle (apical) and to the IHF pial surface (basal); (3) MZG migration via somal translocation to the IHF; (4) MZG differentiation into multipolar astrocytes, including the elaboration of processes that penetrate the fissure; (5) the elimination of the leptomeninges and intervening cells within the fissure and finally; (6) the midline crossing of callosal and hippocampal commissure axons. Most of these steps are disrupted in the absence of DRAXIN, but we propose that the greatest impact on the overall phenotype of BTBR mice comes from its regulation of the earliest steps of IHF remodelling (steps 1-3). More MZG are generated early in development which accumulate at the third ventricle and migrate a shorter distance along the midline pial surface. These cells also have disorganised Nestin-positive radial processes, attached to an enlarged fissure that is filled with more leptomeningeal cells that increase their proliferation at early stages in the BTBR mouse.

Comparing these data with those in DCC/NTN1 mutant mice (see our companion paper), DRAXIN may mediate steps 2-5 above via DCC signalling, but since DCC/NTN1 mutant mice do not show disrupted cellular proliferation of MZG or an expanded IHF, these effects must be mediated via a different pathway, such as by antagonizing canonical WNT signalling to elicit these effects.

We have modified the discussion on pages 19-21 to draw parallels between the phenotype of our BTBR *Draxin* mutation model and *Draxin* knockout mice, as well as to further clarify our model while also discussing other potential mechanisms. We have modified our schemas in figure 8L and 8M to better explain this model.

2. Reviewer 2 asks that the human data be more clearly integrated into the rest of the study. The phenotypes in the mouse are very similar to what is found in humans, and also variable, perhaps related to stochasticity on the mechanisms involved, or to the dependency on other allelic variants. MRI scans of human subjects with a spectrum of CC abnormalities, in addition, shows that commissure abnormalities correlate with midline fusion defects. The human data rely on adult phenotypes and do not relate to Draxin and although it is interesting in itself it gives little insight into the embryonic mechanisms that are so nicely explored in the mouse part of the study. In my opinion, the data should be included but the text in the Results and Discussion to better blend these two aspects.

We undertook a substantive rewrite of the results and discussion to better integrate the findings from human and mouse data. As indicated, we:

– Clarified the rationale for studying humans with partial CCD in the Results section on page 13.

– Rewrote the first paragraph of the discussion to explicitly outline the potential aetiological link between our findings in mice and humans (page 18).

– Now outline the link between the neuroanatomy of human adults with CCD and possible embryonic development scenarios of the corpus callosum in the second discussion paragraph (page 18). Moreover, we also reference our findings from the mice (and previous studies in humans/mice) to strengthen and blend the adult/embryonic findings. Three references were added to the reference list on page 22 (Gloor et al., 1993; Wahl et al., 2009; Tovar-Moll et al., 2007).

– Discuss the implications that the *Draxin* mutation is a driver of these phenotypes in our third discussion paragraph (page 19). We draw on further evidence from other mouse models to support this. We then discuss how the variability in phenotype is likely due to other environmental and genetic influences (one candidate being DCC) and how this relates to what we observe in humans with CC malformations and DCC mutations.

3. The reviewers thought that the use of BTBR x C57 N2 crosses where commissure phenotype is correlated with the Draxin mutation (Figure 5) is a nice illustration of unpicking variable penetrance. However, the description of the findings that prompted you to investigate the role of Draxin in CCD needs to be clearer.Also, It seems that the Draxin deletion does not affect HC formation. However, in the Results section you state that "To investigate how DRAXIN regulates CC and HC formation…". This is confusing. Also, it appears that the effect varies between BTRB mice and the BTRB x C57 cross, but this is not discussed clearly. One solution is to move the Draxin findings to the next to the lat last part of the Results, before the human results.

We thank the reviewers for these suggestions and have made the following additions:

– We introduce *Draxin* as a favourable candidate gene that is located at the chromosome 4 locus identified in our previous study (Jones-Davis et al., 2013: PloS One), since *Draxin* is known to regulate both CC and HC formation in mice (page 4).

– While we showed that HC cross-sectional area was not significantly correlated with SNP allele composition or the *Draxin* mutation allele, HC length was significantly reduced when the *Draxin* mutation was homozygous. We previously showed that IHF remodelling proceeds in a ventral to dorsal direction (Gobius et al., 2016: Cell Reports). Thus, we believe this explains why HC length is a more sensitive measure (than HC area) of abnormal HC development due to the loss of IHF remodelling, which occurs when *Draxin* is mutated. The HC length data was originally in the supplementary figure, but we have now moved it to the main figure 5, and removed the HC area data from the manuscript. This required updating of the figure legend on page 33 and the results text on pages 14-15.

4. As Reviewing Editor: You say that contrary to published evidence on Draxin as a known regulator of axon guidance, "Draxin was not found to regulate axon guidance in this context, but rather impacted the proliferation of astroglia and leptomeninges; two cell populations that are also not yet considered to play a major role in formation of commissures". And "while pathogenic variants in DRAXIN have not yet been reported in human individuals with CCD, a similar effect may underlie the spectrum of phenotypes observed in humans with DCC mutations, since Dcc and Draxin have been demonstrated to interact to determine the severity of CCD in mice (Ahmed et al., 2011)". In the Introduction and/or in the Discussion it would be welcome to mention the relationship of Draxin to DCC as receptors for Netrin , as well as your other submitted study that is on BioRxiv and submitted to eLife on Dcc and Netrin1 regulating intrahemispheric fusion and the midline zipper glia.

We have now introduced *Draxin* as a candidate gene that is known to regulate DCC signalling in the introduction on page 4 and included discussion of our companion paper on page 19 (also see response above). We included our companion paper in the reference list on page 24.

5. Statistics:a. For Figure 1, there is very little information about statistical analysis. For figure 1 C, it needs to be explained why the Welsh test was used instead of a one-way ANOVA. The errors on the bars do not seem to correspond to SEM; this needs to be clarified.

The reviewer is correct that the error bars were indeed representing standard deviation instead of SEM as stated in the figure legend – the error bars have been corrected to show SEM. We agree that a one-way ANOVA would be appropriate as all CCD groups shown in Figure 1C passed the D’Agostino and Pearson test for normality. The figure (Figure 1) has been amended to reflect the minor changes in statistical significance and the figure legend (page 27) has been updated. We have also described more clearly our statistical methods (page 10) and updated our supplementary statistics table (page 2, supplementary file 1).

b. For Figures 3 G and H, if the data are presented in single graphs, it is not clear why unpaired t tests or Mann-Whitney tests were conducted (instead of ANOVAs). Why a non-parametric test was used is not explained.

We chose unpaired t tests or Mann-Whitney tests and not the ANOVA because the data in these graphs are from two groups (control and pCCD), and not from three or more unrelated or independent observation. We have made clear in our methods that we choose our statistical tests based on whether they pass a normality test (page 10).

c. Some of the data are not normally distributed (particularly clear for pink data points in Figure 5a,e,i,m) so it is not appropriate to show standard errors (the SEM bars could simply be removed), a non-parametic Kruskal-Wallis ANOVA has been used which is appropriate.

We choose to show standard error of the mean since it depicts where the mean of the group should fall if the data was sampled again an infinite number of times. We feel this is informative, since even when the data is not normally distributed, it demonstrates where the mean of the population (rather than just one sample) would fall. The calculation of the standard error of the mean is based on the variance in the sample itself and the distribution of this variance does not have to be normally distributed for this to be accurate. We have not made any changes.

Revisions expected in follow-up work:1. You could perform additional experiments on the glial cells in which Draxin is expressed, to better understand mechanism.

We appreciate that this is a central question that remains from our study. There are a number of technical issues that do not make this possible in the short term. Firstly, we have not found a feasible system for ease of manipulation of genes in these glia. We have not found a Cre-expressing transgenic-line that is well expressed in these cells, that is not otherwise expressed in all progenitors in the forebrain. We have not yet been able to efficiently isolate and dissociate these cells into a culture system, since they become reactive and revert to a progenitor state. In slice culture, these glia also become reactive and do not behave in an in vivo state. We have had some success with in utero electroporation of the progenitors of these cells before, but have had issues with efficiency of knockdown of proteins in sufficient time to have an effect. We have been working on developing a better experimental paradigm to study these glia in more depth. We would like to perform single cell sequencing on these glia in order to identify candidate genes that are enriched in MZG to develop cell-specific driver lines for these cells.

2. A central contention of this study is that variable penetrance of the commissure phenotypes in the BTBR x C57 mice stems from an earlier midline fusion phenotype. It would have been useful to discern whether the (embryonic) midline fusion phenotype also showed the same partial penetrance in BTBR x C57 mice, perhaps also correlated with the WT/MUT Draxin alleles (as in Figure 5). This would be a testable prediction of the hypothesis that midline fusion mediates the Draxin phenotype.

We have considered this experiment, but our current data suggest there are unpredictable variables (genetic, cellular, developmental time) that would limit the accuracy of testing this hypothesis. Given the incomplete penetrance of complete and partial CCD in the BTBR x C57 N2 cross, it is not possible to predict on an individual animal level, based on genotype, the final postnatal CC phenotype. It follows that the way in which we would test this hypothesis is to determine whether the ratios of CCD phenotypes detected postnatally (i.e. partial CCD, complete CCD or normal CC) correspond to similar ratios of embryonic cellular phenotypes (i.e. moderately impaired MZG competency, severely impaired, or unimpaired). This presupposes that the measurable cellular phenotypes in embryos can be disaggregated into categorical variables, or clearly delineated quantitative measures, which correspond to adult phenotypes. Given the multifactorial nature of CCD in this model, which could incorporate multiple modulatory genetic factors (as additional QTL have appeared promising but failed to reach significance in either Jones-Davis et al. 2013 or our current study), and multiple cellular factors (MZG, leptomeninges, axon plasticity), we anticipate that such a neat outcome would be unlikely and any measures of cellular phenotypes for a given genotype will fall into a distribution without clear boundaries between expected subgroups. We would anticipate that there would indeed be cut-offs within this cellular phenotype distribution which correspond to final callosal phenotype, but these would need to be inferred from the ratios observed from the final genotypes, and so the logic in identifying these would be circular. A final consideration is that the number of mice required for testing the hypothesis with this number of expected variables is expected to be in the hundreds.